# Atomistic mechanism of coupling between cytosolic sensor domain and selectivity filter in TREK K2P channels

Berke Türkaydin [1,2,5], Marcus Schewe [3,5] ✉, Elena Barbara Riel [3,4], Friederike Schulz [3], Johann Biedermann [1], Thomas Baukrowitz [3] ✉ & Han Sun [1,2] ✉

The two-pore domain potassium ($K_{2P}$) channels TREK-1 and TREK-2 link neuronal excitability to a variety of stimuli including mechanical force, lipids, temperature and phosphorylation. This regulation involves the C-terminus as a polymodal stimulus sensor and the selectivity filter (SF) as channel gate. Using crystallographic up- and down-state structures of TREK-2 as a template for full atomistic molecular dynamics (MD) simulations, we reveal that the SF in down-state undergoes inactivation via conformational changes, while the up-state structure maintains a stable and conductive SF. This suggests an atomistic mechanism for the low channel activity previously assigned to the down state, but not evident from the crystal structure. Furthermore, experimentally by using (de-)phosphorylation mimics and chemically attaching lipid tethers to the proximal C-terminus (pCt), we confirm the hypothesis that moving the pCt towards the membrane induces the up-state. Based on MD simulations, we propose two gating pathways by which movement of the pCt controls the stability (i.e., conductivity) of the filter gate. Together, these findings provide atomistic insights into the SF gating mechanism and the physiological regulation of TREK channels by phosphorylation.

TREK channels belong to the two-pore domain potassium ($K_{2P}$) channels, which are responsible for generating background currents that help maintain the resting membrane potential below the threshold of depolarization[1–3]. They are widespread throughout the cardiovascular, central, and peripheral nervous systems[4–7], playing key roles in diverse physiological processes and diseases such as anesthesia, ischemia, epilepsy, and depression. Loss-of-function mutations in these channels can lead to severe pathologies[8–11]. Structurally, TREK channels can exist as homo- or heterodimeric forms, each subunit comprising four membrane-spanning domains (M1 to M4), an extracellular loop domain (EC1/EC2), two pore-forming domains (P1 and P2) that build a pseudo-tetrameric selectivity filter (SF) (Fig. 1a), and a long

cytoplasmic C-terminus (Ct) connected to the distal M4 helix. While no full-length atomic structure of any $K_{2P}$ channel has been resolved, X-ray structures of TREK-1, TREK-2, and TRAAK have been determined in different N- and C-terminal truncated forms[12–17]. These structures imply flexibility in the long Ct, although AlphaFold2[18] suggested a helical conformation for most of the Ct region (Supplementary Fig. 1). Among these structures, the largest conformational variation occurs in the intracellular section of the pore-lining helices M2 - M4, categorized into two distinct states (Fig. 1a–c): the up-state, where the M4 helix is orientated towards the membrane bilayer, resulting in closure of the intramembrane-facing side fenestrations and the down-state, characterized by M4 orientating away from the inner membrane leaflet,

[1]Leibniz-Forschungsinstitut für Molekulare Pharmakologie (FMP), Berlin, Germany. [2]Insitute of Chemistry, Technical University of Berlin, Berlin, Germany. [3]Institute of Physiology, Kiel University, Kiel, Germany. [4]Department of Anesthesiology, Weill Cornell Medical College, New York, USA. [5]These authors contributed equally: Berke Türkaydin, Marcus Schewe. ✉e-mail: m.schewe@physiologie.uni-kiel.de; t.baukrowitz@physiologie.uni-kiel.de; hsun@fmp-berlin.de

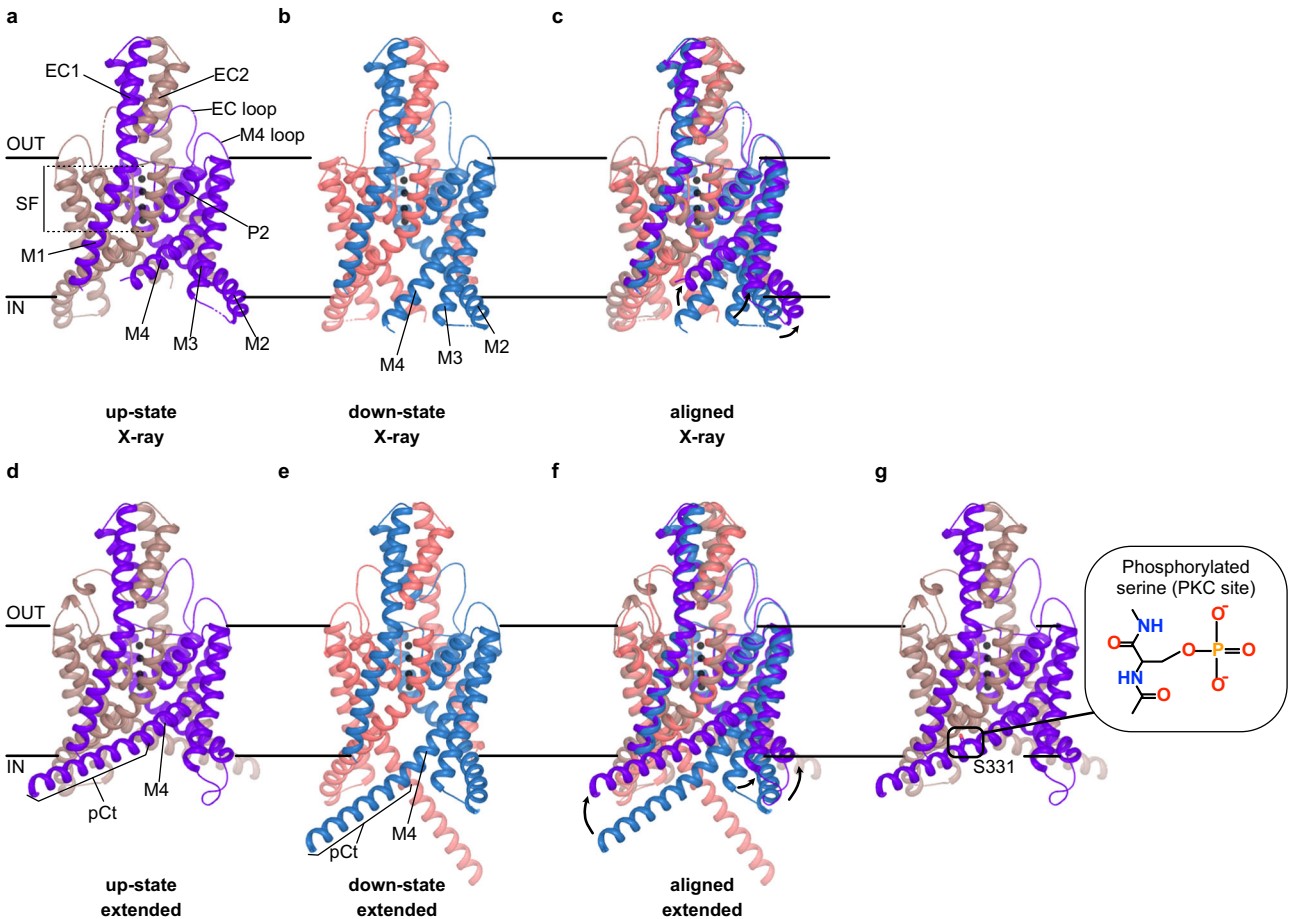

**Fig. 1 | Structures and functional states of TREK-2 K$_{2P}$ channels. a–c** Crystal structures of the (**a**) up-state (PDB ID: 4BW5)[14] and (**b**) down-state (PDB ID: 4XDJ)[14]. TREK-2 channels are shown, along with (**c**) an alignment of both structures. **d–f** The same structures depicted for TREK-2 with a 19-residue extension of the proximal C-terminus (pCt) in their helical conformation (designated as TREK-2*), which served as the starting structure for MD simulations. **g** TREK-2* structure as in (**d**) with S331 in the pCt highlighted. Note, that S331 is phosphorylated in the respective MD simulations.

leading to the opening of the side fenestrations. Due to its larger occupation of membrane volume, the up-state is thought to represent the active state induced by membrane stretching. The down-state is characterized by a high sensitivity to the inhibitors fluoxetine and norfluoxetine (NFx) that bind in the side fenestration and is thought to represent a low activity state[14,19,20]. Alternative concepts have been proposed to explain the differences in activity between the up- and down-states. One concept, derived from the TRAAK crystal structures, suggests a lipid block mechanism[13]. Here, the two crystallographic states show identical and conductive SF. In the down-state, however, an additional electron density below the SF is interpreted as the acyl-chain of a phospholipid, which could potentially block ion permeation. A very recent cryo-electron microscopy (cryo-EM) structure of TREK-1, obtained in the presence of the *n*-dodecyl-β-D-maltoside (DDM) detergent and phosphatidylethanolamine (PE), reports contrasting findings[21]. In the down-state, the positively charged head group of PE binds to the four threonine residues that make up the S4 K$^+$ binding site in TREK-1, rather than an acyl-chain directly blocking the ion permeation pathway.

Another striking feature of K$_{2P}$ channels is their ability to convert various external stimuli into electrical signals. TREK channels, in particular, can be efficiently gated by changes in transmembrane voltage, pH, temperature, mechanical force, polyunsaturated fatty acids like arachidonic acid (AA), bioactive lipids such as PIP$_2$, and phosphorylation[22–26], highlighting them as prototypic polymodal-regulated ion channels. Functional experiments revealed the SF as

the main gate in TREK channels[27–30], but structural changes in the SF have been only revealed at low potassium (K$^+$) concentrations[16]. For sensing and integrating most physicochemical factors, the Ct has been suggested to play a critical role in channel gating. For instance, altering the Ct sequence in the TREK-2 renders it insensitive to pH and fatty acid[31], while truncation of the Ct in TREK-1 channels results in the loss of activation by AA and anesthetics[23,32].

Moreover, TREK channel activity undergoes dynamic regulation by protein kinase A (PKA)-, C (PKC)- and G (PKG)-dependent signaling pathways that are thought to be involved in the development of hyperalgesia under inflammatory conditions and depression via the modulation of serotonin release in raphe nuclei[4,8,33,34]. Phosphorylation at two sites in the Ct has been revealed to be responsible for the inhibition downstream of receptor activation in TREK channels[33,35]. These two sites include S300 in TREK-1 (S331 in TREK-2) for PKC-induced phosphorylation and S333 in TREK-1 (S364 in TREK-2) for PKA-induced phosphorylation. Interestingly, it was shown that (de-)phosphorylation of TREK channels produces a dynamic interconversion between voltage-gated (phosphorylated) and 'leak´-like (dephosphorylated) phenotypes[36].

Direct structural evidence showing allosteric coupling between the proximal Ct (pCt) and the SF gate is currently lacking, as both up- and down-state TREK structures revealed the same SF conformation[14], but several experimental and computational studies have suggested a possible coupling between different conformational states of the M4 helix and the SF gate[15,17,37–40]. However, previous molecular dynamics

(MD) simulations were performed exclusively with the truncated form of TREK-2 lacking the pCt and consequently, the precise allosteric interaction network between these two dynamic hotspot regions remains poorly understood.

In this work, to overcome this limitation, we incorporate a 19-residue pCt for both the up- and down-state structures of TREK-2 (referred to as TREK-2*; see method section) in our MD simulation and conduct an integrated investigation using electrophysiology measurements, encompassing (de-)phosphorylation mimics, state-dependent inhibitors, lipid tethering experiments, and extensive MD simulations. We aim to elucidate (i) the role of the pCt, (ii) pCt coupling to the SF, and finally (iii) the alterations in the filter gate defining TREK channel activity. This approach suggests two distinct structural pathways by which up and down movements of the pCt control the functional status of the SF being either conductive or non-conductive. The non-conductivity of the down-state results from a conformational change at the SF. Overall, our findings provided insights of how different physiological signals like phosphorylation can be sensed and transmitted from the pCt to the SF in TREK $K_{2P}$ channels at the atomistic scale.

## Results

### Modification of the pCt with alkyl-MTS reagents modulates TREK channel activity

To gain mechanistic insight into the regulation of TREK channel activity by the orientation of the pCt, we systematically introduced cysteine residues into the pCt of TREK-1 channels and attached probes of different chemical nature via methanethiosulfonate (MTS)-mediated chemical modification. We hypothesized that introducing a lipophilic probe, such as an aliphatic chain (i.e., a decyl-chain), at sites facing the membrane would foster the interaction of the pCt with the membrane. Likewise, introducing lipophobic (hydrophilic) probes at these sites, such as charged moieties (e.g., MTS-ET$^+$ or MTS-ES$^-$), would destabilize membrane interactions. This approach aims to move the pCt towards or farther away from the membrane and, thereby allowing us to study the impact of this movement on TREK channel function.

First, we measured WT TREK-1, WT TRAAK and TREK-2* channels without modifiable cysteine residues as references. Upon application of 100 μM of decyl-MTS for at least 30 s to the respective channel in excised membrane patches, no change in current amplitudes was recognized (Fig. 2a and Supplementary Fig. 2a, b). Note, that patch integrity and channel regulation were tested with either 1 mM tetra-pentyl-ammonium (TPenA) and a short pH$_i$ activation step. Next, we conducted systematic cysteine scanning mutagenesis of the entire pCt and examined the effect of charged and aliphatic MTS probes. We found that TREK channels carrying cysteine residues in the pCt in a regular interval could be activated robustly with decyl-MTS (Fig. 2a, b and Supplementary Fig. 2c). Remarkably, all hit residues face toward the inner leaflet of the lipid bilayer (Fig. 2b, inlay). The strongest activation was observed for TREK-1 T318C, V322C and F325C, respectively, with full current increases of 33 ± 11-fold, 42 ± 17-fold and 103 ± 20-fold, respectively. Strikingly, the corresponding cysteine mutant channels in TRAAK and TREK-2 could be similarly activated with decyl-MTS (Fig. 2 b, d, e). Furthermore, modification of the hit residues in TREK-1 (T318C, V322C, and F325C) with charged MTS probes like MTS-ES$^-$ and MTS-ET$^+$ resulted in the opposite effect on activity, i.e., channel inhibition (Supplementary Fig. 2d). These results suggest a helical conformation for the pCt in the TREK channel family and that introducing lipid anchors mimicking decyl-groups to the membrane-facing site appears to lock all three TREK/TRAAK channels in a highly active state.

We further investigated this structural transition using MD simulations and in silico attaching either decyl-MTS or MTS-ET$^+$ to the cysteine at position 334 (T334C) of TREK-2* (corresponding to TREK-1 T318). For both decyl-MTS- and MTS-ET$^+$-bound channels, we conducted (i) five runs of 1 μs unbiased MD simulations, starting from the

down-state TREK-2* structure and (ii) five runs of enhanced sampling simulations based on adiabatic bias MD (ABMD) approach[41] to characterize the required force to move the system from the down- to up-state. In the unbiased simulations of the decyl-MTS-modified channel, we observed the insertion of the decyl-chain chain into the lipid bilayer (Fig. 2f), while membrane embedding of the -ET$^+$ moiety was never observed (Fig. 2g). This differential behavior was supported by calculating the distance between the ligands and the lipid bilayer in the z-direction (Fig. 2h and Supplementary Fig. 3). These results suggest that attaching a decyl-group to the pCt promotes movement of the pCt towards the membrane. The results from ABMD simulations further supported this finding, showing that the required force to move pCt/M4 in the decyl-MTS-attached system is considerably less than in the MTS-ET$^+$ one (Fig. 2i and Supplementary Fig. 4). Whether this induces a state similar to the crystallographic up-state was further tested functionally by using the state-sensitive inhibitor NFx, as NFx only binds to the down-state. To this end, we measured NFx inhibition for the TREK-1 F325C mutant channels before and after decyl-MTS modification. Before modification, F325C mutant channels showed a similar high sensitivity to NFx as WT channels, indicating a large fraction of the channels in the NFx-sensitive down-state. In contrast, decyl-MTS modification caused a dramatic drop in NFx sensitivity (Fig. 2c) indicating that a large fraction of channels now exist in an up-state like conformation.

### The difference in ion conductivity is a result of the conformational change in the SF of TREK-2*

The results of the modification of the pCt with alkyl-MTS reagents have established that the upward movement of the pCt induces an active channel state that likely resembles the crystallographic up-state. Previously, we and others have shown that TREK channel activity is controlled by a gate residing in the SF, but the crystallographic up- and down-states of TREK have identical SF structures, which argues against the contribution of the SF in TREK channel gating. However, the crystallographic conditions could have altered the energetics of the SF, thereby obscuring its intrinsic properties. Therefore, we performed MD simulations of K$^+$ permeation (RMSD of the simulations shown in Supplementary Fig. 5) in the up- and down-state of TREK-2 with an extended pCt (TREK-2*). All simulation runs of the up-state TREK-2* were conductive at positive voltages (around +200 mV) throughout the entire 1 μs MD runs (Fig. 3a and Supplementary Table 1). In a typical 1 μs trajectory under +200 mV, the channel remained conductive with an average of five K$^+$ permeation events, corresponding to a conductance of 4 pS. In contrast to the up-state, no K$^+$ conduction was observed in the down-state TREK-2* simulations (Fig. 3a and Supplementary Table 1). Therefore, our results reveal a strong correlation between ion conductivity and the conformational state of pCt/M4. It is noteworthy that we observed very few ion conduction events under negative transmembrane potential for the up-state and no ion conduction for the down-state (Supplementary Fig. 6). Whether this result is related to voltage dependency of the TREK channel as previously revealed[27], or it is an artifact associated with force field imperfections as demonstrated in other K$^+$ channels using similar methods[42,43], future systematic studies are warranted.

Similar to the previous simulations of the down-state TREK-2 channel by Aryal et al.[44], we also observed penetration of single or multiple lipids from the lateral side fenestration during the simulations (indicated by red arrows in Fig. 3c, d and Supplementary Fig. 7). This took place mostly in the down-state simulations, but was also seen in some of the up-state runs (Supplementary Figs. 7, 8) and did not obviously block ion conduction.

Analysis of one- and two-dimensional ion occupancies suggested significant differences in the preferred K$^+$ binding sites in the SF region between the up- and down-state (Fig. 3b and Supplementary Fig. 9) simulations of TREK-2*. While ion binding at sites S0 - S5 was almost

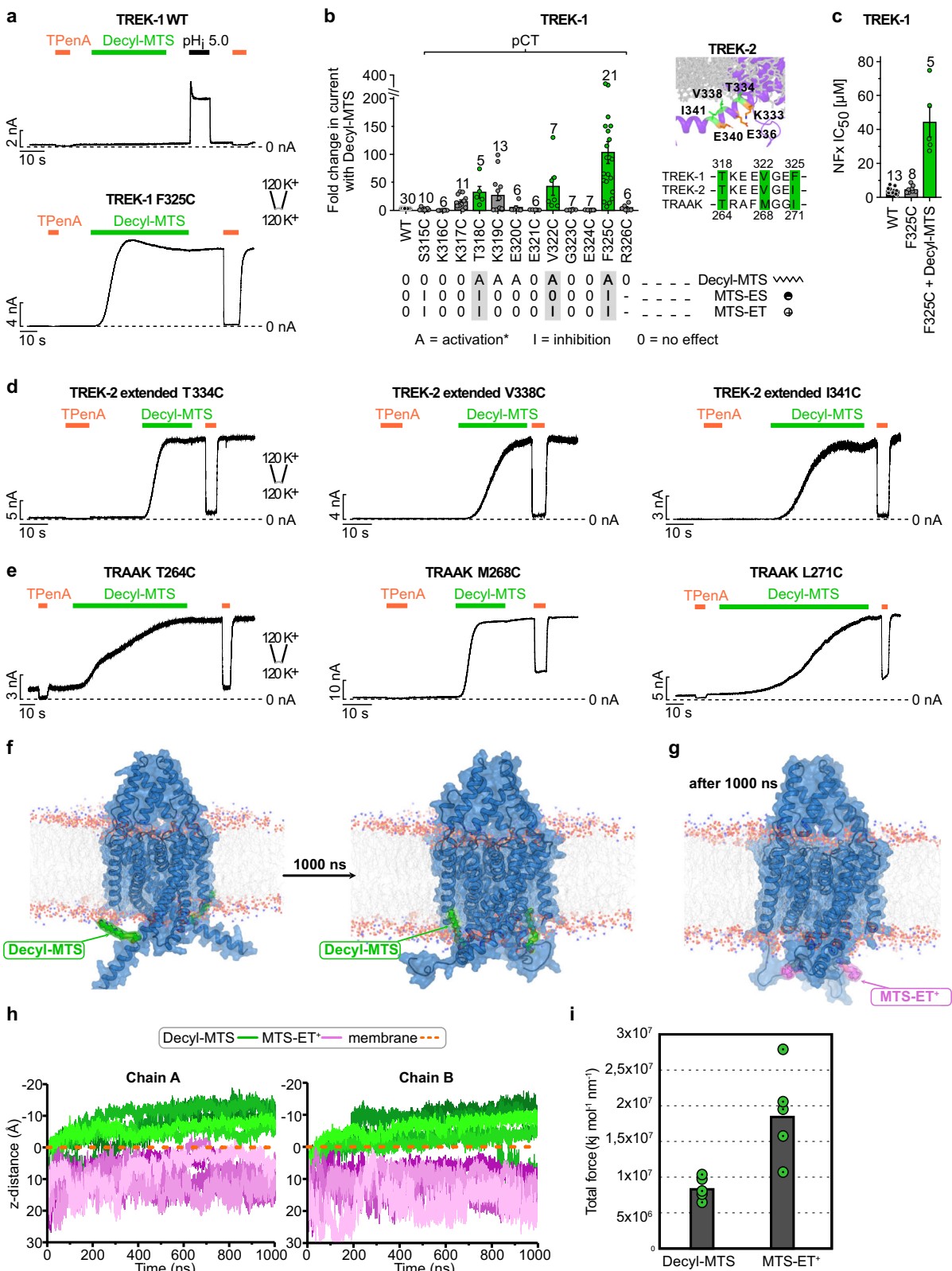

equally populated during the conductive up-state simulations, ion binding at S1 and S4 was abolished in the simulations of down-state TREK-2 (Fig. 3c, d). Due to significant differences in ion occupancy between up- and down-state simulations, we further compared the SF backbone conformation in these simulations (Supplementary Fig. 10). Notable differences were identified for Y175 and F284 around the S1 K$^+$ site (Fig. 3e). While the crystallographic SF conformation remained

stable throughout the entire conductive up-state simulations for all five runs, in the non-conductive down-state simulations, the backbone carbonyl of Y175 and F284 flipped away from the ion conduction pathway during all five trajectories and remained stable for the remainder of the simulation time (Fig. 3e, f and Supplementary Figs. 11, 12). Building upon previous MD studies that demonstrated carbonyl flipping in the SF can impede ion conduction in K$^+$ channels[16,39,45],

**Fig. 2 | Membrane tethering of the pCt with cysteine modifying probes activates TREK channels. a** Example recordings of WT and F325C mutant TREK-1 channels in a symmetric $K^+$ gradient at pH 7.4 at + 40 mV showing no effect upon application of 100 μM decyl-MTS for WT and robust activation for F325C mutant channels. **b** Systematic screening of cysteine mutants in the pCt region of TREK-1 by recordings as in (**a**) reveal mutants that are activated (A) with decyl-MTS. Note, that side chains of mutant channels that are strongly activated (green sticks) point to the inner leaflet of the lipid bilayer phase (gray sticks) as highlighted for TREK-2* in the inlay. Mutant TREK-1 channels with robust decyl-MTS activation show an inhibition (I) upon MTS-ET$^+$ application. **c** Apparent affinities (IC$_{50}$ values) of NFx for WT, F325C mutant, and decyl-MTS-activated F325C mutant TREK-1 channels. In panels (**b**) and (**c**), values are given as mean ± s.e.m with number (n) of individual experiments indicated in the figure. **d, e** Example recordings as in (**a**) of TREK-1 homologous mutant channels with the strongest decyl-MTS response in TREK-2* (**d**) and TRAAK (**e**). Note, that TREK-2* for functional recordings corresponds to the construct used in MD. **f** Representative initial (left) and final (right) snapshots of decyl-MTS-modified TREK-2* from a 1 μs unbiased MD simulation. **g** A representative final snapshot of MTS-ET$^+$-modified TREK-2* from a 1 μs unbiased simulation. **h** The distance (D$_z$ component of the vector) between a chosen atom of the ligand (C12 for decyl-MTS, N2 for MTS-ET$^+$) and the membrane plotted against time over 5 runs of unbiased 1 μs simulations, conducted under around + 200 mV. The membrane was defined as a plane by selecting the center of geometry of phosphorus atoms located in the cytosolic part of the POPC lipid bilayer. **i** The average total force required for the down-to-up transition of M4/pCt for a decyl-MTS and MTSET$^+$ attached TREK-2 channel, respectively, derived from 5 runs of 10 ns ABMD simulations. Green circles represent the total force for single runs.

we propose that the ion non-conductivity in the down-state simulations resulted from the carbonyl flipping of Y175 and F284 in TREK-2*, leading to a strong reduction in the ion occupancy at S1 and S4.

## Phosphorylation at the PKC site in the pCt promotes the up-to-down transition

Having established that up- and down-movement of the pCt controls TREK channel activity through allosteric coupling with the SF, we explored whether a physiologically relevant stimulus employs the same mechanism. We focused here on the regulation of TREK channels by PKC and PKA, as both enzymes exert a strong inhibitory effect via phosphorylation sites at the Ct. The PKC site is located in the pCt (S300 in mTREK-1 and S326 in rTREK-2), whereas the PKA site is located at a more distal site in the Ct (S333 in mTREK-1 and S359 in rTREK-2)[33,35,46]. Recently, it was shown that in the process of receptor- and kinase-mediated modulation of TREK channels a sequential phosphorylation is responsible for the inhibitory effect on channel activity[33] with PKA phosphorylation promoting PKC phosphorylation. In the same study it was shown that alanine substitutions at both sites (S300A/S333A) mimicking a permanently dephosphorylated state enhance TREK-1 channel currents, while a double aspartate mutation (S300D/S333D) mimicking permanently phosphorylated channels drastically reduced TREK currents.

To investigate the mechanistic basis for phosphorylation regulation of TREK channels we first tested if the (de-)phosphorylated states with enhanced or diminished activity can be assigned to either the up- or down-state conformation of TREK utilizing as before the state-sensitive inhibitor NFx (Fig. 4a)[14,20,47]. We measured TREK-1 channel currents in response to 800 ms voltage ramps (− 80 mV to + 80 mV) in excised membrane patches under asymmetrical $K^+$ conditions (Fig. 4b). TREK-1 WT is inhibited by NFx with an apparent affinity of ~ 7 μM (IC$_{50}$ = 7 ± 1 μM, n = 7) (Fig. 4b,d,e). In agreement with previous studies, the double alanine substitution in hTREK-1 (S315A/S348A; equivalent to S300A/S333A) and hTREK-2 (S331A/S364A; equivalent to S326A/S359A) resulted in largely increased currents (Supplementary Fig. 13a). Notably, NFx sensitivity of these currents were strongly reduced and even high concentrations produced incomplete inhibition (IC$_{50}$ = 249 ± 149 μM, n = 5 for TREK-1 and IC$_{50}$ = 24 ± 4 μM, n = 9 for TREK-2) (Fig. 4d–f and Supplementary Fig. 13b). In contrast, the phosphorylation mimicking substitutions S315D/S348D in TREK-1 and S331D/S364D in TREK-2, respectively, resulted in smaller currents (Supplementary Fig. 13a) and slightly increased NFx sensitivities compared to WT (IC$_{50}$ = 2 ± 1 μM, n = 6 for TREK-1 and IC$_{50}$ = 1 ± 1 μM, n = 7 for TREK-2) (Fig. 4d–f and Supplementary Fig. 13c). Further, the PKC site (S315) appeared to dominate the overall impact of phosphorylation on TREK activity as the (de-)phosphorylation mimicking alanine (S315A) or aspartate (S315D) substitutions in TREK-1 and TREK-2 had a similar impact on NFx sensitivity and current amplitude as seen for the respective double substitutions (Fig. 4c–f and Supplementary Fig. 13d, e).

These results indicated that mutations at the PKC site mimicking dephosphorylation induce a down-to-up state transition in TREK channels, while phosphorylation mimics retain the channels in a down-state conformation. This also implies that TREK channels at least in our expression system are strongly phosphorylated without external kinase stimulation as introducing the phosphomimic substitutions only slightly increases the NFx sensitivity.

We further explored the impact of phosphorylation at the PKC site with MD simulations, introducing a phosphate group in silico at residue S331 of TREK-2* in the up-state (Fig. 1g). We performed seven runs of 3 μs MD simulations at 300–320 K (details in the method section). Notably, in 3 out of 7 simulations, we observed a complete transition from the up- to down-state (Fig. 4g), where the C-terminus could adopt an even deeper down-state compared to the crystallographic down-state structure of TREK-2 (Fig. 4g, h). This result is partially in line with a previous computational study on TREK channels[39]. From the simulations it is also evident that phosphorylation at S331 disrupts the hydrophobic interaction between S331 at the pCt and F202 at M2 from the opposite subunit (Fig. 4k). Following the reduction of this hydrophobic interaction, we observed in some of the simulation runs that a lipid occupied a site between M2 and M4 (Fig. 4l). Importantly, during the simulations where the complete up-to-down transition occurred, we again observed conformational changes at the S1 binding site in the SF, where backbone carbonyl of Y175, F284 orientated away from the ion conduction pathway (Fig. 4i and Supplementary Fig. 14). These results, showing a complete transition from up- to down-state followed by the SF inactivation at the S1 binding site, provide additional evidence of the coupling between the C-terminus and SF.

In addition, we conducted well-tempered metadynamics simulations[48] for the up- and down-state, respectively, to further explore the energetics of SF inactivation. Here, the Ψ dihedral angle changes of F284 and Y175 in the SF (Fig. 4m) were defined as two collective variables (CVs). The free-energy surface (FES) profiles unveiled three energy minima for the up-state (Fig. 4n): (i) up-I corresponds to the conductive state, resembling the X-ray structure; (ii) up-II represents a single flipped conformer of F284; (iii) up-III, the lowest energy conformation, corresponds to a doubly flipped conformation of F284 and Y175. In contrast, for the down-state, we identified four energy minima. Down-I and down-II both correspond to single-flipped conformations of Y175, while down-IV represents the single-flipped conformation of F284. Down-III corresponds to a doubly flipped conformation of both F284 and Y1175. Notably, no energy minimum for the conductive state was observed in the down-state. This finding underscores a distinct energy difference between the conductive and non-conductive conformations in up- and down-state simulations. Collectively, our functional and computational data suggest that phosphorylation at the PKC site promoted a downward movement of the pCt leading to a conformational state resembling the down-state in the X-ray structure, where simultaneously the SF inactivation occurred.

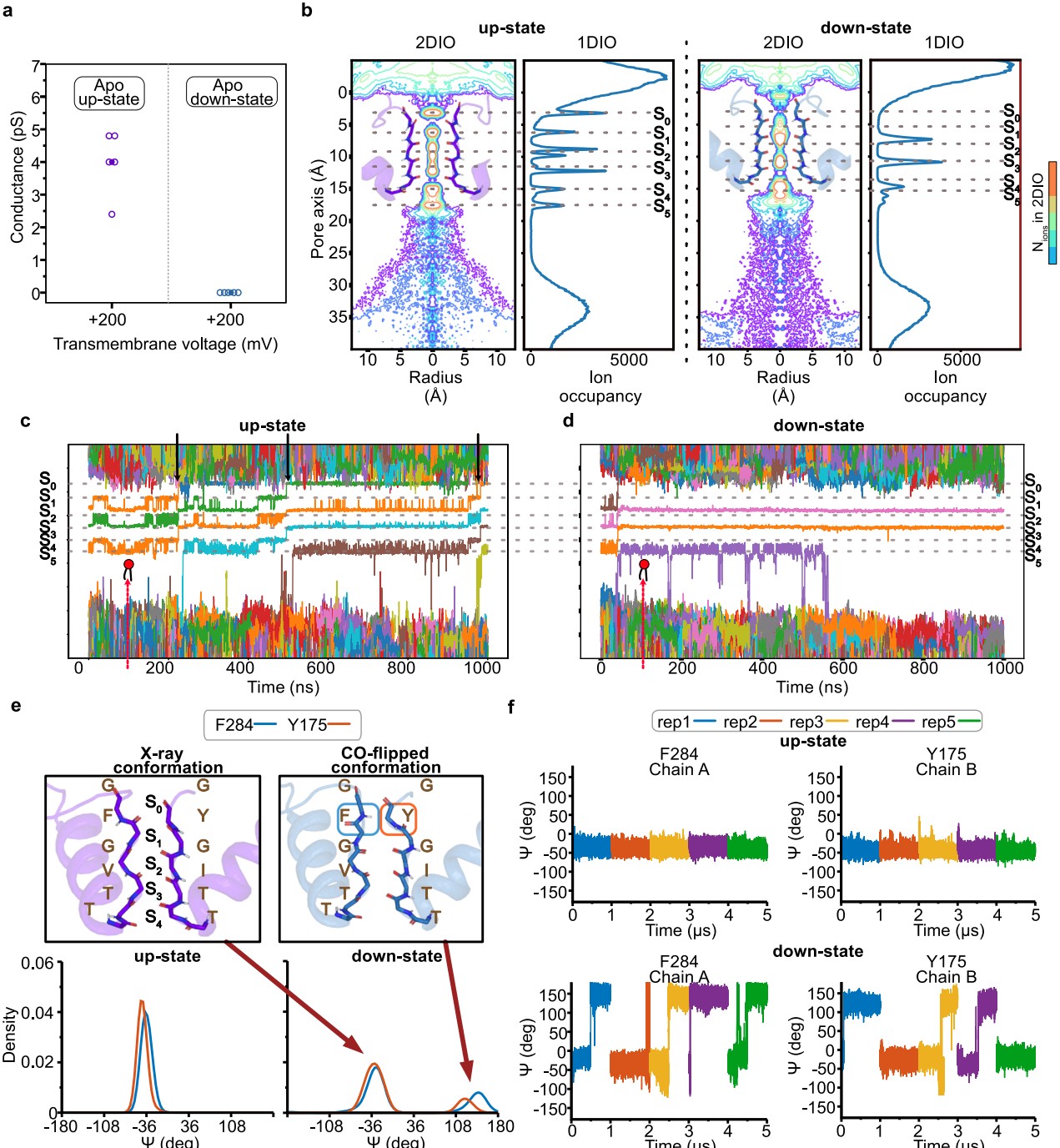

**Fig. 3 | Conductance, SF ion occupancy and conformational changes determined from TREK-2* simulations. a** Ion conductance derived from apo TREK-2* simulations in its up- (PDB ID: 4BW5)[14] and down-state (PDB ID: 4XDJ)[14]. Filled and empty circles represent mean and individual ion conductance, respectively. **b** One- and two-dimensional ion occupancy profiles within the SF for combined up-state and non-conductive down-state TREK-2* simulations. The radial area of the pore was defined and ions passing along the pore axis ($D_z$) were calculated from the simulations. The occupancy of ions was normalized per 0.001 Å³ per 1 μs based on the volume change along the radius. The center of mass of the SF backbone atoms was located on the 7 Å point along the pore axis. Ion binding sites along the pore axis were indicated with dashed lines. **c** Traces of ions passing through the SF during a representative 1 μs up-state and (**d**) down-state TREK-2* simulation. Black arrows indicate the respective ion permeation events, while red arrows indicate lipid penetration from the lateral side fenestration. **e** Distribution of Psi angle (ψ) for F284 and Y175 from the combined up-state (left) and down-state (right) simulations. **f** Psi angles (ψ) variations of F284 in chain A and Y175 in chain B over time in the up-state (top) and down-state (bottom) of TREK-2*. Individual simulation runs were carried out with AMBER99sb[59] for 1 μs under a transmembrane voltage of around + 200 mV.

## Interaction network analysis revealed the coupling mechanism between SF gate and pCt in TREK-2*

To gain further insights into the coupling pathway between the pCt/ M4 and the SF, we conducted an extensive interaction network analysis. We identified intra- and inter-chain contacts between residues within a 5 Å radius during up- and down-state simulations, respectively. By subtracting the population of down-state contacts from the up-state ones, we constructed a contact difference map to highlight key

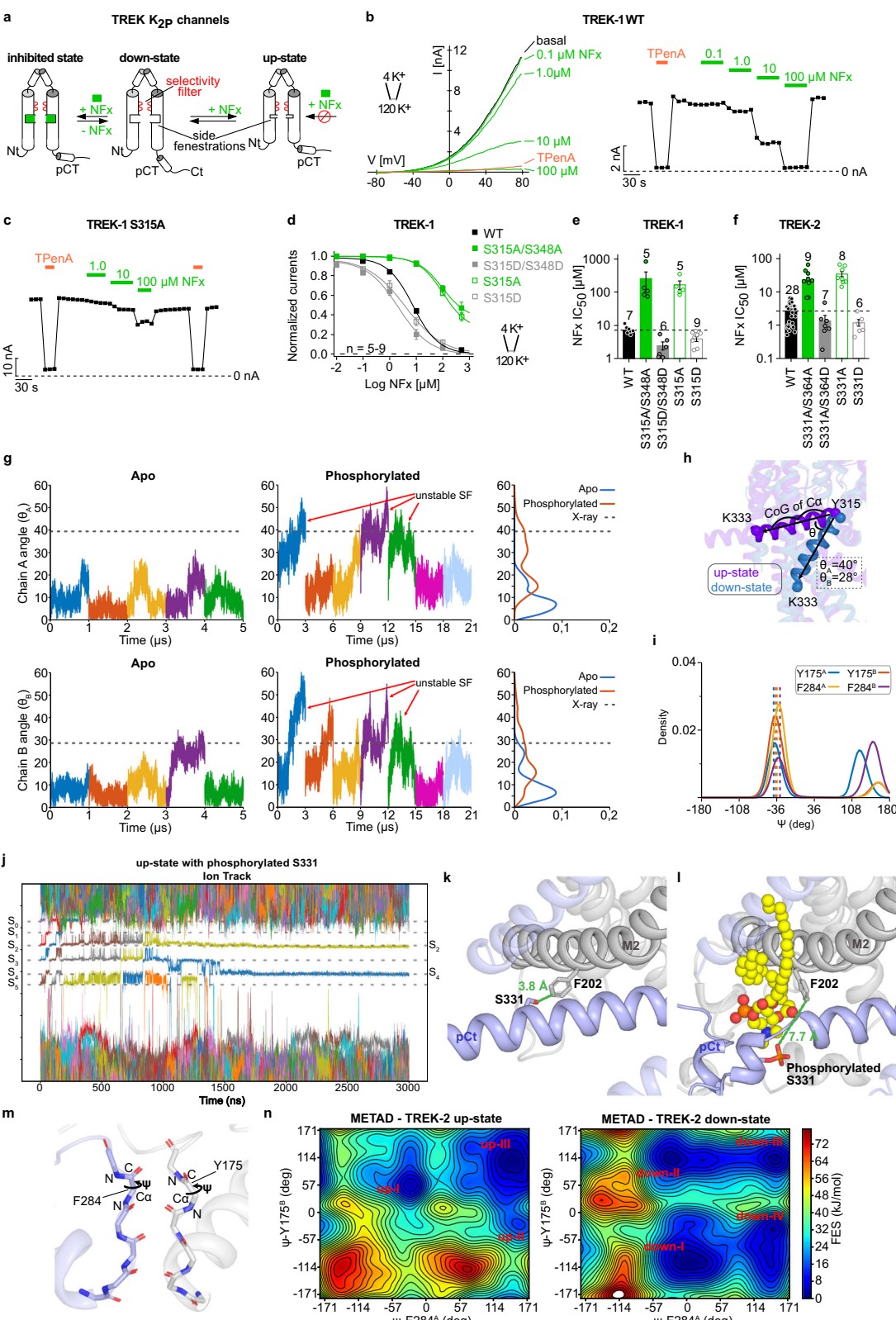

disparities. In total, we identified 86 contact differences (> 0.25 probability), comprising 22 inter-chain (transmembrane helices of both channel subunits) contacts and 64 intra-chain (transmembrane helices of the same channel subunit) interactions (Supplementary Table 2).

Analyzing the contact differences, first of all, we observed substantial variations between up- and down-states at the interfaces of M2

and pCt/M4 (Fig. 5a and Supplementary Fig. 15). Specifically, we identified five hydrophobic contacts predominantly present in up-state simulations between the pCt/M4 and M2 helices from opposing subunits. One of these involved P198 from the M2 helix, engaged in three of the five identified contacts (with F316, L320, and I323 from M4). S331 - F202 and F316 - I197 constituted two other key inter-chain

**Fig. 4 | Phosphorylation-mediated state transition of TREK channels. a** Cartoon depicting the gating model and state dependency of NFx inhibition in TREK K$_{2P}$ channels. **b** Example recording of WT TREK-1 channels in an asymmetric K$^+$ gradient at pH 7.4 in a voltage range from −80 to +80 mV showing the dose-dependent inhibition with the indicated concentrations NFx (left) and the time course of block analyzed at +40 mV (right). **c** Same recording and analysis as in (**b**) for S315A TREK-1 channels. **d** Dose-response curves of NFx inhibition for WT and mutant TREK-1 channels as indicated analyzed from recordings as in (**b**). **e, f** NFx IC$_{50}$ values from dose-response curves as in (**d**) for WT and mutant TREK-1 (**e**) and TREK-2 (**f**) K$_{2P}$ channels. In panels (**d–f**), values are given as mean ± s.e.m with the number (**n**) of individual experiments indicated in the figure. **g** Time course of the θ-angle for M4 in the apo and phosphorylated TREK-2* simulations starting from the up-state structure, shown for the chain A and B. Histograms of θ-angle distribution are included on the right site. θ-angle of the up-state and down-state X-ray structures

was used as a reference and shown as a dashed line. Simulations with unstable SF are indicated with a red arrow. **h** θ-angle in the up- and down-state X-ray structures. **i** Distribution of Psi angle (ψ) for F284 and Y175 from three replicates of 3 μs phosphorylated simulations with unstable SF. **j** Traces of ions passing through the SF during a representative 3 μs phosphorylated TREK-2* simulation. The temperature of the system is increased from 300 K to 320 K after 1000 ns. **k** The distance between S331 and F202 was illustrated for the unphosphorylated TREK-2* in the up-state and (**l**) for a phosphorylated TREK-2* after 1 μs simulation. **m** Representation of selected collective variables (Ψ of F284 and Y175) for well-tempered metadynamics simulations. **n** Two-dimensional free energy surface (FES) derived from a 500 ns well-tempered metadynamics simulation for the up-state and down-state TREK-2*, respectively. The FES contoured in steps of 4 kJ/mol up to 80 kJ/mol. Three energy minima were identified for the up-state, while four energy minima were identified for the down-state.

contacts distinguishing the up- and down-states. In contrast, the key inter-chain interactions between M4 and M2 were largely disrupted in the down-state simulations, being replaced by new interactions between M4, M2 and M3 within the same subunit, such as W326 - R237, R237 - E223 and R237 - K224 (Fig. 5a). We computed the distance distribution of the inter-chain interaction P198 - L320 and intra-chain interaction L211 - A318 throughout the simulations. In the up-state simulations, the distance between P198 and L320 remained small, even slightly smaller compared to the crystallographic state in one subunit (Supplementary Fig. 16a). However, in the down-state simulations, this interaction became highly asymmetric and was exclusively abolished in one subunit. This trend was inverted for the intra-chain interaction L211 - A318 (Supplementary Fig. 16b). Notably, this result is inconsistent with the finding of the phosphorylated TREK-2* simulations, where the disruption of the hydrophobic interaction between S331 and F202 was found to be critical for inducing the down-state.

Triggered by the differences in interactions between M2, M3, and M4, the dynamics of the pCt/M4 and the SF were further coupled through two distinct pathways. The first pathway involves the M2 helix and the P2 pore-forming domain (Fig. 5, primarily inter-chain interactions), where interactions such as I197 (M2) - F316 (M4), F316 (M4) - T280 (P2), Y273 (P2) - F284 (SF2), T278 (P2) - Y175 (SF1), Y273 (P2) and P180 (P1-M2 linker) were predominantly observed in the up-state simulations. Conversely, in the non-conductive down-state simulations, a series of new interactions emerged, including P180 (M2) - E99 (M1) and Y273 (P2) - C189 (M2). In the second pathway, the coupling of the pCt/M4 and the SF was mediated by the M4 loop, which connects SF2 and the upper M4 helix (Fig. 5). Several differences were identified between the M4 loop and its interacting counterparts. For instance, the interaction between Y175 (SF1) and T278 (P2) stabilized the conductive SF conformation in the up-state simulations, but this interaction diminished and was replaced by the Y175 (SF1) - V288 and A289 (M4 loop) interactions. Another example is the dominant interaction between Y273 (P2) and D286 (M4 loop) in the up-state, which was replaced by the D286 (M4 loop) - N141 (EC2) interaction in the down-state simulations.

In addition, based on the phosphorylated simulations where a full transition of the C-terminus from the up- to the down-state occurred, we performed a new interaction network analysis using PSNtools[49]. This approach allows us to understand allosteric communication between distal sites by computing the shortest communication pathways through the interaction network. As shown in Fig. 5b, in the initial 1 μs when the C-terminus adopted an up-state, communications between the C-terminus and the SF are mainly mediated through the M4-P2-SF pathway, whereas in the subsequent 2 μs where the C-terminus transits to the down-state and SF inactivation occurred, this network shifts to the M4 - M4 loop - SF pathway. Overall, the interaction network analysis using two different approaches unveiled the same interaction preferences and pathways for the up- and down-states.

## Discussion

TREK K$_{2P}$ channels constitute background K$^+$ channels that are susceptible to regulation via G-protein-mediated pathways and protein kinases[33,35]. Phosphorylation at the Ct by PKG, PKA, and PKC inhibits TREK channels. These kinases target specific Ct phosphorylation sites, with cross-influence on each other's activity. Notably, PKC possesses a phosphorylation site in the pCt domain. Previous studies have also suggested the essential involvement of the pCt in PIP$_2$ activation[50], pH regulation, and mechanosensitivity[51]. In this study, based on electrophysiological measurements of (de-)phosphorylation mimics at the PKC site, we were able to demonstrate that phosphorylation strongly affects the up- and down-state conformational equilibrium. These results have been further supported by atomistic MD simulations, revealing the potential for a complete conformational transition from the up- to down-state upon phosphorylation at the PKC site and reduced pCt-membrane interaction.

Modifying the pCt with alkyl-MTS probes of varying hydrophobicity yielded contrasting channel activities and conformational equilibria between up- and down-state, further confirming the crucial regulatory role of the pCt domain in TREK channels. While the hydrophobic interaction of the pCt/M4 and the membrane was reinforced by embedding decyl-MTS into the membrane, MD simulations of the MTS-ET$^+$ variant did not demonstrate the same side-chain insertion, leading to a reduced hydrophobic interaction between pCt and membrane.

Microsecond MD simulations of apo up- and down-state TREK-2 with extended pCt revealed a notable difference in the conductance of two distinct conformational states, particularly regarding the outward flow of K$^+$. Debates persist regarding whether the less-conductive down-state arises from a lipid pore block[13] or conformational changes in the gating region. While we observed lipid penetration from the lateral side fenestration in several down-state TREK-2* simulations, similar to previous simulations[44,52], we also observed the same in some of our up-state simulations without significant obstruction of channel conductance (Supplementary Figs. 7, 8).

MD simulations revealed considerable differences in S1 K$^+$ occupancy between up- and down-state TREK-2*. Non-conductive down-state simulations showed reduced S1 ion density due to the reorientation of the backbone carbonyls of Y175 and F284. Although no conformational difference was observed in the SF of the original X-ray structures between the up- and down-state TREK-2, we noticed that continuous ion density at the S1 - S4 sites was present only in the up-state structure. In contrast, the down-state X-ray structure lacks ion density at site S1[14]. The absence of ion occupation at the S1 site in the down-state X-ray structure reflects the dynamic nature of this ion binding site. In line with that, another structural study revealed substantial conformational changes relayed to the S1 and S2 K$^+$ sites in the X-ray structures of TREK-1 at low K$^+$ concentrations[16]. Similarly, cryo-electron microscopy (cryo-EM) structures of the non-conductive state of TASK-2 K$_{2P}$ channels showed conformational changes of F101 in SF1

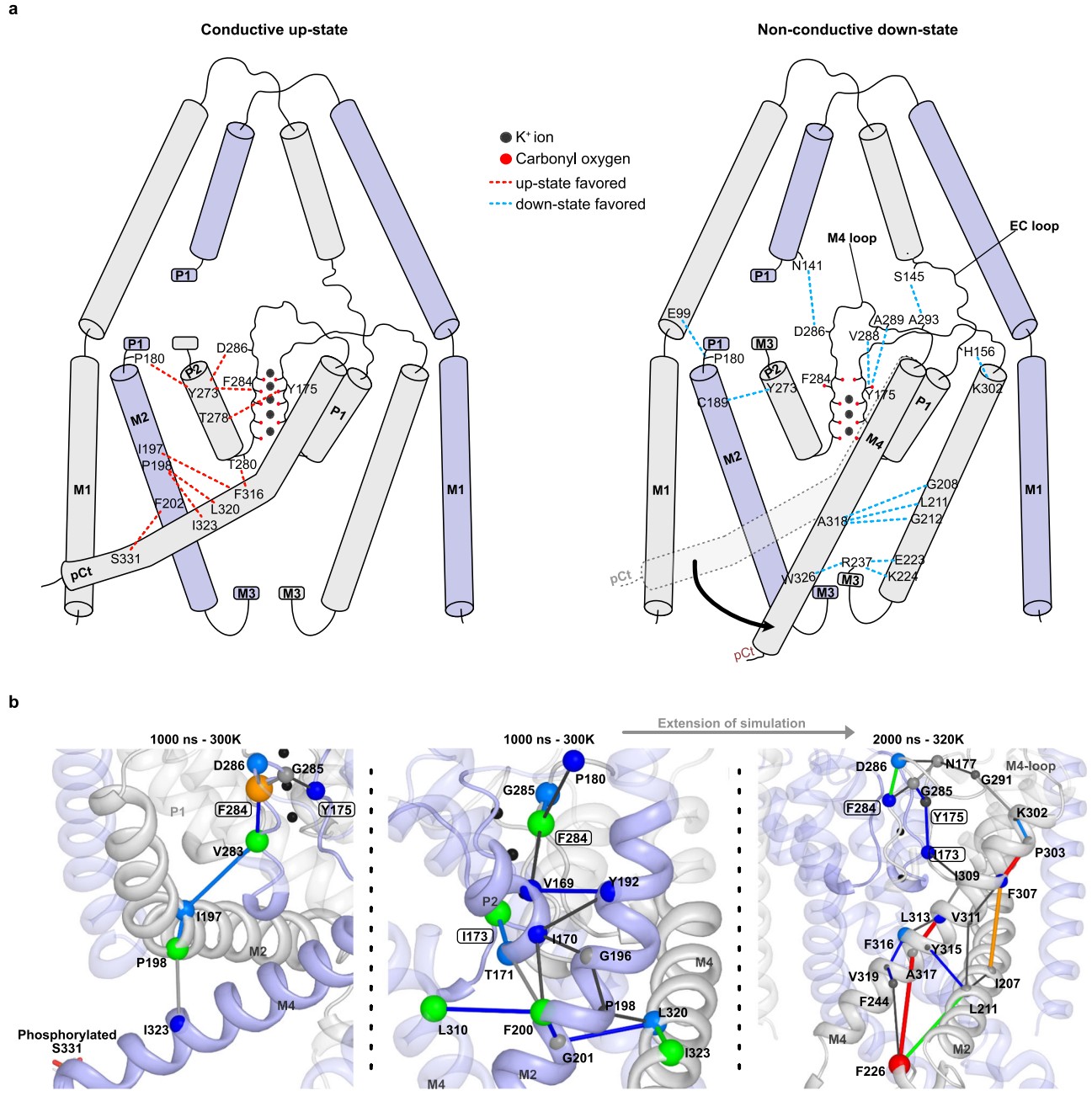

**Fig. 5 | Coupling mechanism between pCt/M4 and SF gate. a** pCt/M4 dynamics trigger different inter- and intra-chain M2 - M4 interactions that are further coupled to the SF gate via two different pathways: (i) M4 → P2 → SF; (ii) M4 → M4 loop/EC loop → SF. Favorable interactions in the up-state are depicted with red dashed lines, while blue dashed lines represent favorable interactions in the down-state. **b** Protein structure network analysis using PSNTool[49] on a 3 μs phosphorylated TREK-2* simulation confirmed that the up-state (the initial 1 μs) is mainly stabilized by the M4 → P2 → SF pathway, whereas in the following 2 μs during the transition between up- and down-state and SF inactivation, the interaction pathway shifted to M4 → M4 loop → SF.

and F206 in SF2 at the S1 ion binding site[53]. These observations suggest a consensus mechanism for C-type gating in the SF within the $K_{2P}$ channel family, primarily focused on the S1 site. Interestingly, a recent MD work by Pettini et al. also revealed a similar depletion of ion binding at S1 in the hERG (human Ether-a-go-go Related Gene) channel[45]. A comparison of the SF sequences between most $K_{2P}$ channels (except TWIK channels) and canonical $K^+$ channels revealed a major difference: in K2P channels, one (in SF2) or both (in SF1 and SF2) tyrosine (Tyr) residues adjacent to the S1 $K^+$ site are replaced by phenylalanine (Phe) (Supplementary Fig. 17). This sequence alteration leads to a reduction in hydrogen bonding interactions between the SF and the pore-forming helix, potentially decreasing the stability of the

SF at the upper part. The hERG channel, which also possesses a Phe at the homologous position in the filter, exhibit side-chain shifts in the cryo-EM structures[54] and ion depletion at S1 in the MD simulations[45], were suggested to be associated with fast inactivation. Nevertheless, we propose that the configuration without ions at the S1 site, coupled with the carbonyl reorientation of Y175 and F284, does not represent the true SF conformation of inactivated TREK-2* channels. Previously, from the gating charge analysis, we proposed the inactive filter to be in an 'ion-depleted' state[27]. The full inactivation of the SF was not observed in the current study most probably due to the limited time scale of the simulations, as well as the choice of using additive force field - without considering the polarization effect - may contribute to

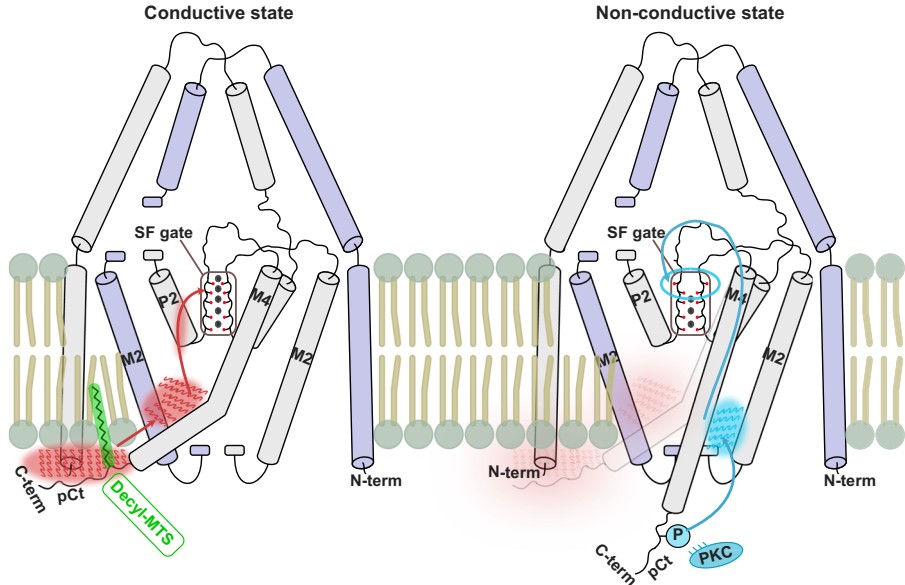

**Fig. 6 | Schematic illustration of TREK-2 K$_{2P}$ channel gating and modulation.**
The conductive state is stabilized by the strong interaction between pCt and the membrane, promoting inter-chain M4 - M2 interactions. These interactions are coupled with the SF gate through M4-P2 pathway, thereby stabilizing the conductive state of the SF. Disruption of the interactions between pCt and the lipid bilayer leads to the emergence of new interactions between the M4 and M2 of the same subunit, resulting in conformational changes of the SF mediated by the M4 loop that leads the channel to a non-conductive state. Decyl-MTS (green) enhances the interactions between pCt and the membrane. In contrast, phosphorylation of S331 at the pCt (blue) destabilizes the interactions between pCt and the membrane, leading to a non-conductive state of the channel.

additional inaccuracy. Nonetheless, based on the functional and simulation evidence discussed above, we propose an allosteric mechanism that links the dynamics of pCt/M4 with C-type gating in the SF, as illustrated in Fig. 6.

Firstly, substantial variations in the M4 and M2 interaction interfaces were revealed between the up- and down-states. In the conductive up-state simulations, inter-chain interactions between M4 and M2 dominated, whereas, in the non-conductive down-state simulations, this interaction network was diminished and replaced by intra-chain interactions of M4, M2, and M3. Phosphorylation at S331 disrupts the hydrophobic interaction between pCt/M4 and M2 from the opposite subunits, thereby promoting the down-state. Triggered by these differences in the interaction network, two key interaction pathways, that mediate the coupling between M4 and the SF have been further identified: (i) M4 → M4 loop/EC loop → SF, where interactions between the M4 - loop and surrounding residues (e. g. in EC loops) facilitate the conformational transition in the upper part of the SF. This pathway dominates when the C-terminus is in down-state. These findings from MD simulations align well with previous functional and simulation studies, which suggested that M4-loop dynamics play a crucial role in C-type gating[16]. (ii) M4/M2 → P2 → SF, where the cross-talk between M4 dynamics and the SF gate is mediated by the interactions between M2, M4, and P2 (Fig. 5). This pathway dominates when the C-terminus is in up-state. It should be noted that the interaction between the P2 and M4 helix is primarily mediated by the interaction between F316 (M4) and T280 (P2), an interaction that was predominant only in the up-state simulations. Interestingly, the significance of this interaction in mediating the coupling between the M4 helix and the SF had already been proposed in previous simulations of the K$_{2P}$ channel TRAAK[55].

In conclusion, using an integrated approach, combining electrophysiology including systematic cysteine scanning mutagenesis, lipid tethering and blocker experiments, molecular dynamics simulations and interaction network analysis, we have highlighted the essential role of the pCt in TREK channel gating. We also revealed how the interaction between the pCt domain and the membrane influences the conformational equilibrium between the up- and down-state of TREK channels. Furthermore, MD simulations proposed an allosteric mechanism at the atomistic scale, delineating two pathways of the energetic coupling between the cytosolic sensing domain and the SF. These pathways are in good agreement with previous functional and mutational studies. However, it is important to note that while MD simulations at higher temperatures suggested the possibility for a complete conformational transition from the up- to down-state upon phosphorylation at the PKC site, our enhanced sampling simulations did not achieve sufficient convergence to quantitatively characterize free energy differences between these two states. This issue is most probably due to the high flexibility of the C-terminus (Supplementary Figs. 20, 21). Therefore, future mutational work is necessary to further validate the coupling pathways proposed in the current study. In addition, more advanced enhanced sampling methodologies should be explored shortly to better assess the conformational transition between the up- and down-states in a more quantitative manner.

## Methods

### MD-based ion permeation simulations of up- and down-states of TREK-2*

We started MD simulations from the high-resolution X-ray structure of up- (PDB ID: 4BW5)[14] and down-state (PDB ID: 4XDJ)[14] TREK-2 channels, respectively. The missing pCt in each subunit was manually extended using the PyMOL builder option[56]. To preserve the helical conformation, the helix option was selected during the addition of the amino acid sequence. The pCt was extended by adding 19 residues ($^{333}$TKEEVGEIKAHAAEWKANV) based on the TREK-2 construct[57] (TREK-2*). We added N-methyl amide and acetyl caps to the N and C termini using the PyMOL builder option. SWISS-MODEL[58] was used to build missing loops (residues 149 to 154 and 229 to 235).

We performed MD simulations with AMBER99sb[59] force field. The insertion of the TREK-2 channel into a POPC was performed with the GROMACS internal embedding function. The concentration of KCl was

600 mM in simulations. Improved ion parameters[60] and lipid parameters[61] were employed in the simulations. The SPC/E water model[62] was used in simulations.

All unbiased MD simulations were performed with the GROMACS software (2019.3 and 2021.2)[63]. Short-ranged electrostatics and van der Waals interactions were truncated at 1.0 nm. Long-range electrostatics were calculated with the Particle-Mesh Ewald summation[64]. Temperature and pressure coupling were treated with the V-rescale[65] scheme and the Parrinello-Rahman barostat[66], respectively. The temperature was set to 300 K and the pressure to 1 bar. Fluctuations of the periodic cell were only allowed in the $z$-direction, normal to the membrane surface, keeping the density of the membrane unchanged. Verlet cut-off scheme was used for neighbor searching and pair interactions[67]. All bonds were constrained with the Linear Constraint Solver (LINCS) algorithm[68]. We used virtual sites for hydrogens to decrease the computational cost of apo- and phosphorylated TREK-2 simulations. These simulations were performed with an integration time step of 4 fs. All other TREK-2 simulations with covalently bound MTS reagents used an integration time step of 2 fs.

After the channel was embedded into POPC with ions, the system was energy minimized and equilibrated. The energy minimization was done with GROMACS "steepest descent" algorithm with a maximum energy value of 100 kJ mol$^{-1}$ to make sure to minimize the energy of the system as much as possible. After the minimization, the system was equilibrated for 10 ns with a position restraint by a force constant of 1000 kJ mol$^{-1}$ nm$^{-2}$ in $x$, $y$, and $z$-direction on the backbone atoms of the protein followed by 20 ns without position restraints on the backbone atoms of the protein. During the complete equilibration, the isothermal–isobaric (NPT) ensemble was used.

The simulations were initiated with a four-ion configuration (S1 - S4) in the SF for all setups, consistent with the ion configuration in the X-ray structure of the TREK channel in the up-state (PDB ID: 4BW5)[14]. We used the same ion configuration in the down-state simulations to avoid any influence on ion conductance caused by different initial ion configurations.

A copy of the equilibrated TREK-2 system was prepared and two identical copies were stacked on top of each other in the $z$-direction to construct a CompEL setup. Another energy minimization was performed for the new double-bilayer system to prevent crashes of molecules located in the intersection point of the two systems. An ion difference of 2 was introduced between the two compartments separated by lipid bilayers to introduce a transmembrane potential of about ± 200 mV (Supplementary Fig. 18). Additional particle interchange algorithm (deterministic protocol) was employed during MD simulations to keep the number of ions in every compartment constant alchemically[69].

### MD simulations of MTS-attached TREK-2*

For the in-silico attachment of the MTS reagents, methyl sulfonyl groups of decyl-MTS and MTS-ET$^+$ were removed to represent solely the side-chain structure, respectively. A cysteine residue was fused by a disulfide bridge to the trimmed reagent. Cysteine residue was capped with N-methyl amide and acetyl capping groups. The geometry of the structure was optimized and the electrostatic potential was calculated at the Hartree-Fock/6-31 G* level using Gaussian16[70]. The generalized amber force field (GAFF)[71] topology of the structure was generated with the Antechamber software[72] using partial charges from the preceding quantum mechanics calculations according to the restrained electrostatic potential (RESP) approach[73]. After the geometry optimization, caps and cysteine were removed from the structure. The remaining side chain was fused by a disulfide bridge to the T334C mutant located in pCt. The disulfide bridges were generated at a distance of 2.05 Å.

Energy minimization, equilibration, and 5 runs of 1 μs unbiased MD simulations of decyl-MTS- and MTS-ET$^+$-attached systems were performed in the same way as described above for the up- and down-state of TREK-2*.

The required force for the transition from the down- to up-state in the presence of different MTS reagents was determined by performing adiabatic bias MD (ABMD)[41] simulations in GROMACS 2022.5 patched with PLUMED 2.9.0[74,75]. ABMD is an enhanced sampling approach that dynamically adjusts the harmonic potential as the system approaches the target configuration along the collective variable. This adjustment effectively enhances the sampling of the system towards the desired value of the collective variable. By continuously adapting the potential landscape, ABMD optimizes the exploration of relevant regions in the system's configuration space, leading to more accurate and efficient simulations. The collective variable is defined as the distance between the center of masses of Cα atoms from Y315 to K333 (representing M4) and from I194 to G206 (representing M2), for the decyl-MTS- and MTSET$^+$-attached systems, respectively. 5 runs of 10 ns ABMD simulations were performed for the decyl-MTS- and MTSET$^+$-attached systems, respectively. The force constant for the harmonic potential was set to 5000 kJ/mol/nm$^2$ and the target value for the collective variable was set to 0.5 nm. The square force was used to calculate the total force by summing the forces applied along the trajectory.

### MD simulations of phosphorylated TREK-2*

For the in-silico preparation of the phosphorylated serine residue in the TREK-2* in up-state structure, the hydrogen of the hydroxyl group was removed, and a phosphate group was added to the oxygen using the builder option of PyMOL. The hydrogen atom in the phosphate group was removed to maintain the phosphoserine at a -2 charge. Additional force field parameters for phosphorylated serine were adopted from previous studies[76].

We conducted 7 runs of 3 μs unbiased simulation of the phosphorylated system, starting from the up-state. In the replica 1 and 2, two 1 μs were performed at 300 K, followed by additional 2 μs simulations at 320 K, resulting in a total of 3 μs trajectories. For replicas 3–7, 3 μs simulations were exclusively performed at 320 K. Parameter and other conditions were consistent with the unbiased up- and down-state simulations described above.

To characterize the free energy difference between conductive and non-conductive SF conformations, we conducted 500 ns well-tempered metadynamics simulations[48] of up- and down-state TREK-2*, respectively. The psi angles of F284 and Y175 residue were selected as two collective variables for the system. Simulations were carried out using PLUMED 2.9.0 implemented in GROMACS 2022.5. The biasing potential was added every 250 steps, with the width and height of the Gaussian hills set to 0.25 and 1.2 kJ/mol, respectively, with a bias factor of 8. Frequent visits of different conductive and non-conductive conformations were observed during the 500 ns simulation runs (Supplementary Fig. 19).

In addition, we attempted to simulate the up-to-down transition in the C-terminus using well-tempered metadynamics, experimenting with different CVs and Gaussian heights. However, we encountered challenges in achieving satisfactory convergence, primarily due to the high flexibility of the C-terminus, as suggested by the secondary structural analysis using the Gromacs dssp tool (Supplementary Figs. 20, 21).

### Analysis of the MD data

All trajectories were analyzed with GROMACS tools and Python using MDAnalysis[77] together with NumPy[78] matplotlib[79] and SciPy[80]. Distances were calculated with GROMACS tool $gmx$ $dist$ and dihedral angles were calculated with $gmx$ $rama$. Ion occupancies and the trace of ions were estimated with Python using MDAnalysis by aligning different simulation setups using the center of mass of the backbone atoms of the SF residues as a reference point. Presented data of ion occupancies were derived from the combined trajectories of five

independent replicates from + 200 mV in the up-state (Fig. 3b), as well as five independent replicates from + 200 mV in the down-state. Presented data of dihedral angle distributions were calculated using the kernel density estimation of combined trajectories of five independent replicates from + 200 mV (Fig. 3e).

To calculate the θ-angle of the M4 helix, we designated the Cα atom of the Y315 residue as the initial point of the vector. The second point of the vector was defined as the geometric center of the Cα atoms of residues located between Y315 and K333. Using a combination of MDAnalysis and NumPy, we computed the vector difference of M4 for each frame relative to the reference X-ray structure.

The interaction network analysis was performed using the combined trajectories of up-state apo TREK-2 at + 200 mV, each comprising a 1 μs simulation run. The same was repeated for the combined trajectories of non-conductive down-state TREK-2 trajectories. A pair of residues were considered to be in contact if the distance between any selected two side-chain heavy atoms from the residues was less than 5 Å (CZ of Arg, CG of His, NZ of Lys, CG of Asp, CD of Glu, OG of Ser, CB of Thr, CG of Asn, CD of Gln, SG of Cys, CA of Gly, CG of Pro, CB of Ala, CB of Val, CB of Ile, CG of Leu, CE of Met, CZ of Phe, OH of Tyr, CE2 of Trp). We selected only one heavy side chain atom to decrease the computational cost. The contacts in each frame of combined trajectories were calculated with Python using MDAnalysis. A difference contact map was derived by subtracting down-state contacts from the up-state ones identified from the respective simulations.

The phosphorylated up-state simulations of TREK-2* were analyzed using PSNtools[49]. This analysis aimed to explore the structural communication and allostery of the protein by calculating the correlations of atomic fluctuations across the trajectories. The foundation of PSN analysis lies in applying graph theory to the structures of protein and nucleic acid, where a graph is defined by vertices (nodes) and the connections (edges) between them. In PSNs, amino acid residues are depicted as nodes, and the edges between them indicate the intensity of non-covalent interactions among the residues. To understand the coupling between pCt and SF, the shortest path method of PSNtools was used. This method begins by identifying all possible communication paths between pairs of nodes and then refines these paths to ensure that atomic movements along them are significantly correlated. Only the shortest paths involving residues that demonstrate a correlation higher than a certain cutoff, set at 0.8 nm for networks from MD trajectories, are retained. These refined paths are subsequently used to construct the main meta path, incorporating the most frequently observed connections.

All simulation details were summarized in Supplementary Tables 3 and 4. Molecular visualizations were rendered using PyMOL[56]. Figures including Figs. 5a, 6 and Supplementary Figs. 8a, 17 were generated using Affinity Designer.

## Functional electrophysiology
**Molecular biology.** In this study, the coding sequences of human $K_{2P}2.1$ TREK-1 (Genbank accession number: NM_172042), rat $K_{2P}2.1$ TREK-1 (NM_172041.2), human $K_{2P}10.1$ TREK-2 (NM_021161) and human $K_{2P}4.1$ TRAAK (AF247042) were used. For K⁺ channel constructs expressed in *Xenopus laevis* oocytes the respective K⁺ channel subtype coding sequences were subcloned into the dual-purpose vector pFAW. The TREK-2 construct with the extended pCt (TREK-2*) was truncated from WT and includes amino acids 67 - 355. All mutant channels (point mutations) were obtained by site-directed mutagenesis with custom oligonucleotides. Vector DNA was linearized with NheI or MluI and cRNA synthesized in vitro using the SP6 or T7 AmpliCap Max High Yield Message Maker Kit (Cellscript, USA) and stored at −20 °C (for frequent use) and −80 °C (for long term storage).

**Inside-out patch-clamp measurements.** *Xenopus laevis* oocytes were surgically removed from anesthetized adult females, treated with type II collagenase (Sigma-Aldrich/Merck, Germany) and manually defolliculated. A solution containing the K⁺ channel-specific cRNA at a desired concentration was injected into Dumont stage V - VI oocytes and subsequently incubated at 17 °C in a solution containing (mM): 54 NaCl, 30 KCl, 2.4 NaHCO₃, 0.82 MgSO₄ x 7 H₂O, 0.41 CaCl₂, 0.33 Ca(NO₃)₂ x 4 H₂O and 7.5 TRIS (pH 7.4 adjusted with NaOH/HCl) for 1 - 7 days before use. Patch-clamp recordings in an inside-out configuration under voltage-clamp conditions were performed at room temperature (22 - 24 °C). Patch pipettes were made from thick-walled borosilicate glass GB 200TF-8P (Science Products, Germany), had resistances of 0.2 - 0.5 MΩ (tip diameter of 10 - 25 μm) and filled with a pipette solution (in mM): 120 KCl, 10 HEPES and 3.6 CaCl₂ (pH 7.4 adjusted with KOH/HCl). Intracellular bath solutions and compounds were applied to the cytoplasmic side of excised patches for the various K⁺ channels via a gravity flow multi-barrel pipette system. The intracellular solution had the following composition (in mM): 120 KCl, 10 HEPES, 2 EGTA, and 1 Pyrophosphate (pH adjusted with KOH/HCl). Currents were recorded with an EPC10 amplifier (HEKA electronics, Germany) and sampled at 10 kHz or higher and filtered with 3 kHz (-3 dB) or higher as appropriate for the sampling rate.

**Animals.** For the electrophysiological experiments in this study, oocytes were obtained from adult (2 - 8 years), sexually mature, female Xenopus laevis frogs (Nasco). Inves7ga7ons in this study conform to the guide for the Care and Use of Laboratory Animals (NIH Publica7on 85-23). Twenty-five animals were used to isolate oocytes. Experiments using Xenopus frogs were approved by the local ethics commission of the Ministerium für Landwirtschaft, ländliche Räume, Europa und Verbraucherschutz (IX 555 – 106759/2023 (33-6/23 V)).

**Drugs.** Tetra-pentyl-ammonium chloride (TPenA) was purchased from Merck (Sigma-Aldrich), prepared as 100 mM stocks in intracellular recording solution, stored at −20 °C and diluted to a final concentration in intracellular recording solution before measurements. Decyl-MethaneThioSulfonate (Decyl-MTS), (2-(trimethylammonium)ethyl) MethaneThioSulfonate bromide (MTS-ET⁺) and Sodium (2-sulfonatoethyl)MethaneThioSulfonate (MTS-ES⁻) were purchased from Toronto Research Chemicals. MTS-ET⁺ was directly dissolved to the desired concentration of 1 mM in the intracellular recording solution before each experiment. MTS-ET⁺ was used immediately after dilution for maximally 5 min. Decyl-MTS and MTS-ET⁻ were prepared as 10 - 100 mM stocks in DMSO and diluted to the desired concentration before the measurements. Norfluoxetine hydrochloride (NFx) was purchased from Cayman Chemicals and prepared as 10 mM stocks in methanol.

## Data acquisition and statistics
Data analysis and statistics for functional electrophysiology were done using Fitmaster (HEKA electronics, version: v2x73.5, Germany), Microsoft Excel 2021 (Microsoft Corporation, USA), and Igor Pro 9 software (WaveMetrics Inc., USA).

Recorded currents were analyzed at voltage defined in the respective figure legend from stable membrane patches. Patch integrity was tested with a $K_{2P}$ channel-specific blocker (1 mM TPenA).

The fold current increase (fold activation) of a ligand (decyl-MTS) was calculated from the following equation:

$$Fold\ activation\ (FA) = \frac{I_{activated}}{I_{basal}} \tag{1}$$

with $I_{activated}$ represents the stable current level in the presence of 100 μM decyl-MTS and $I_{basal}$ the measured current before application of the MTS probe.

The half-maximal concentration-inhibition relationship of a ligand (NFx) was obtained using a Hill-fit for dose-response

curves as depicted below:

$$\% \, inhibition = \frac{I_{base} + (-I_{base})}{\left\{ 1 + \left[ \frac{x_{1/2}}{x} \right]^{rate} \right\}} \qquad (2)$$

with base and max are the currents in the absence and presence of a respective ligand, x is the concentration of the ligand, $x_{1/2}$ is the ligand concentration at which the inhibitory effect is half-maximal, and the rate is the Hill coefficient.

Data from individual measurements were normalized and fitted independently to facilitate averaging. Throughout the manuscript, all values are presented as mean ± s.e.m. with n indicating the number of individual executed experiments. Error bars in all figures represent s.e.m. values with numbers (n) above indicating the definite number of executed experiments. A Shapiro-Wilk test or Kolmogorow-Smirnow test was used to determine whether measurements were normally distributed.

Image processing and figure design were done using Affinity Designer, Igor Pro 9 (64 bit) (WaveMetrics, Inc., USA), and Canvas X Draw (Version 20 Build 544) (ACD Systems, Canada).

### Reporting summary

Further information on research design is available in the Nature Portfolio Reporting Summary linked to this article.

## Data availability

Data supporting the findings of this manuscript are available from the corresponding authors upon request. The trajectory data together with simulation parameters and initial structure of trajectories have been deposited in Zenodo under accession code https://doi.org/10.5281/zenodo.11352833.

## Code availability

Molecular dynamics simulation data were generated using the GROMACS 2019.5/2022.5 and PLUMED 2.9. All trajectories were analyzed with GROMACS tools and Python using MDAnalysis together with NumPy matplotlib and SciPy.

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

## Acknowledgements

This work was funded by the Deutsche Forschungsgemeinschaft (DFG) RU2518 DynIon (P1 to M.S. and T.B., P3 to H.S.), under Germany´s Excellence Strategy—EXC 2008/1–390540038 "Unifying Systems in Catalysis" and the Leibniz-Forschungsinstitut für Molekulare Pharmakologie (FMP) (to H.S.). The MD simulations were performed with resources provided by the North-German Supercomputing Alliance (HLRN). We would like to acknowledge the help and support of Dr. Tillmann Utesch as well as Prof. Bert de Groot and Dr. Songhwan Hwang for helpful discussions.

## Author contributions

H.S., M.S., and T.B. designed and directed the project with contributions from all authors. B.T. conducted the MD simulations, while the data was analyzed and interpreted by B.T., J.B., and H.S. M.S., E.B.R., and F.S. performed patch-clamp experiments and M.S., E.B.R., F.S., and T.B. analyzed the data. B.T. and M.S. prepared figures. H.S., M.S., B.T., and T.B. wrote the manuscript with contributions from all authors.

## Funding

## Competing interests

The authors declare no competing interests.
