## [Peer Review File · Nature Communications]

Reviewers' Comments:

Reviewer #1:

Remarks to the Author:

This is an interesting study of the molecular factors modulating the activity of the TREK-2 K^{2P} potassium channels.

Structural studies have revealed a up- and down-state conformation TREK-2. The up-state is believed to represent the highly active state induced by membrane stretching. But what is going on is unclear. In both states, the selectivity filter is essentially unchanged. The only structural changes in the selectivity filter have been only revealed at low potassium (K⁺) concentrations. The down-state is believed to be inactive (non-conductive). In some structures, a detergent or lipid molecules is observed blocking the pore, though it is unclear whether this is an artifact of purification (like with the nAChRs channels). The stated goal of the study is to elucidate the role of the proximal C-terminus, its coupling to the selectivity filter, and the gating mechanism from the filter.

The results about the effect of phosphorylation promoting the up to down transition appear reasonable. In effect, the up and down states appear to be governed by changes in solvation free energy. Hydrophobic interactions promote the position in the membrane (up) and phosphorylation promote the position in the solution (down). The timescale of the simulations is probably too short to reveal the entire transition, but they move in the correct direction. This also points to a main weakness in the usage of MD here. It is all based on relatively mind-less brute-force simulations, with no attempt to characterize free energy differences using enhanced sampling methodologies (meta dynamics, umbrella sampling, etc). It is very nice to see that a piece of an helix is moving in the right direction, but it would be more convincing to show that the free energy landscape has been altered quantitatively. Furthermore, the analysis of the short-range Coulomb and Lennard-Jones energies between the pCt and the membrane with regards to phosphorylation is pretty meaningless. The hydrophobic effect is not a LJ interaction, and direct electrostatic interactions are all shielded by solvation (dielectric effect in the language of continuum electrostatics).

The least convincing aspect of the work concerns the ion conduction simulations. Simulations of the up-state at +200 mV displayed five K⁺ conduction events on average. But simulations at -200 mV did not conduct ions. Simulations of the down-state also did not conduct. Does one expect the IV curve to display such non-linear rectifying behavior for the active channel? Overall, these are very small and relatively inconsistent differences in behavior between these various systems. As a point of comparison, the dilated C-type inactivated Shaker channel from Swartz and co-workers (Tan et al, Science Advances 2022) conducts K⁺ in MD simulations unless the structure is restrained. The pathway along the dilated conformation is not occluded, so the free energy barrier must be just larger to stop conduction. One big concern is whether MD force fields are accurate enough to capture such a subtle inactivation mechanism. The main difference between the conductive and non-conductive selectivity filter is attributed to ion occupancy in the side S1 and the reorientation of the backbone carbonyls of Y175 and F284. But here again, the actual free energy difference dictating the conformational change is not characterized. Ultimately, one does not say why are these carbonyls affected by the rest of the structure. Could one predict one mutation that would prevent this change in the selectivity filter?

In final analysis, while the combination of experiments and computations is highly commendable, I find that the main results are suggestive, but not entirely compelling. Some trends can be explained qualitatively (effect of phosphorylation for example) but would benefit from more systematic enhanced sampling methodologies. Other results such as the conduction or lack of conduction of the channel in the up or down states display small differences of uncertain statistical significance and proposed molecular explanations have to rely on subtle effects that may be highly dependent on the force fields and sampling.

Minor points:

A criticism concerns the style. Many sentences use "functional electrophysiology" to mean

"experimental" measurements, and "computational electrophysiology" to mean MD simulations. I had to re-read carefully several statements to make sure if it was about an experimental fact or a MD result. Personally I dislike the expression computational electrophysiology, which I find needlessly confusing. It is nice to coin cute names, but this does not help the clarity of the text. In addition, there are unjustified emphatic affirmations throughout the text. For example page 12: "Our findings so far have established that the upward movement of the pCt induces a highly active channel state that likely resembles the crystallographic up-state". Highly active? How? Or page 17: "Microsecond MD simulations of apo up- and down-state TREK-2 with extended pCt revealed a striking difference in the conductance of two distinct conformational states" So now the difference of 5 ions conducted has become "striking"?

The transmembrane potential was implemented via the charge imbalance across a double bilayer system. The simpler and more direct alternative would have been to implement the transmembrane potential via an external linear electric field. With the charge imbalance methodology, the transmembrane potential has to be tightly monitored to show that it is indeed the desired target value. The issue is even more critical in the case of non-equilibrium ion conduction. It is stated "An ion difference of 2 was introduced between the two compartments separated by lipid bilayers to introduce a transmembrane potential of about ± 200 mV. Additional particle interchange algorithm (deterministic protocol) were employed during MD simulations to keep the number of ions in every compartment constant alchemically." This needs to be better documented.

In some of the simulation runs a lipid occupied a site between M2 and M4 (Fig. 3L). Are there two lipids associated with the channel (in a symmetric manner) or only one lipid is observed?

Reviewer #2:

Remarks to the Author:

The gating mechanism of two-pore domain potassium channels is still far from being completely understood. This study reports novel information about how the C-terminal of helix M1 regulates the transition between the down- and up-states of the channel and how the movements of this C-terminal modify the properties of the selectivity filter. These results are interesting contributions to the current understanding of how phosphorylation regulates TREK channels. The methods are thoughtfully planned, with experiments and molecular dynamics simulations that coherently support the conclusions of the manuscript. In summary, I believe that this is a well-designed and clearly written work that significantly contributes to the field. Below, I have included some minor comments.

- The Methods section does not report the initial configuration of ions and water molecules inside the selectivity filter and cavity of the channel. Given the low number of conduction events (null in some atomic models), it is not possible to exclude that some results might depend on the initial ion configuration inside the channel. Therefore, I suggest the authors to explicitly discuss this issue in the manuscript, clearly define how and why the atomic models were initialized with respect to the ions inside the channel, and confirm if this initialization was consistent across all replica.

In agreement with the discussion in the final part of page 18, the depletion of ions from S1 was recently reported in simulations of the hERG channel by my group in *J. Chem. Inf. Model.* 2023, 63, 251–258, where we formulated a similar hypothesis that this event might represent an initial step of inactivation.

- These are a few typos or unclear sentences that I noted: page 3 "membrane leaflet with opening", "recently cryo-electron"; page 4 "activation AA and anesthetics"; page 9 "appeared to dominated"; page 17 "interaction differences in interactions".

Reviewer #3:

Remarks to the Author:

This study by Türkyaydin and colleagues combines functional studies of TREK channels with MD simulations and proposes an explanation for the molecular mechanism of phosphorylation-mediated regulation of channel activity, and a general framework for how down-to-up transitions of TM4 is coupled to activation of the SF gate. Using elegant MTS-modification, phospho-mimetic mutagenesis, and norfluoxetine sensitivity experiment - and comparison with MD simulations - the authors convincingly show that membrane anchoring of TM4 increases channel opening which is a plausible explanation for the inverse inhibitory effect of phosphorylation of the extended M4 helix.

The molecular dynamics approaches are mostly outside of my expertise and so I cannot comment on the technical elements. However, the interaction network analysis supports previous models that upwards movement of TM4 is associated with opening of the SF gate and highlights previously identified and potential novel residues responsible.

The paper is very well written, experiments are beautifully performed and displayed, and results provide interesting advances which improve our understanding of TREK channel gating. However, the model could be tested, validated, and strengthened significantly by the following - and this is an excellent opportunity to clarify some important outstanding questions in the field:

* Data suggests phosphorylation of TREK channels drives conversion to low activity "down"-state. This is an interesting and important finding, which would help to explain TREK channel regulation in vivo. The (de)phospho-mimetic experiments are illuminating, but a telling killer experiment showing direct phosphorylation effects is lacking. For example, can the authors excise patches from oocytes in which the PKC pathway has been inhibited, and then demonstrate that NFX sensitivity is markedly reduced, as would be expected for their model? This would provide strong support for a phosphorylation-induced shift in up vs down conformations.

* The interaction network analysis provides possible explanations for how TM4 movement is transduced to SF gate opening. As mentioned, this analysis highlights a number of residues which have previously been shown to significantly affect channel activity when mutated (e.g. W326, R237 in Dong et al., 2015). The authors are well placed to determine whether mutation of these residues, in silico, destabilizes the down-state and to show how this might promote stabilization of the conductive SF.

* The interaction analysis proposes novel functionally important residues. Patch clamp analysis of key mutants would significantly strengthen the proposed model. For example, does mutation of C189, the G208/L211/G212 cluster, T278, or T280 affect channel opening, up-state stability, phosphorylation-regulation or NFX sensitivity?

Major comments:

* Despite the interaction network analysis, it was not clear to me why the conductive SF conformation is disfavored in the down conformation. Is there a temporal correlation with CO-flipping and key interactions elsewhere in the channel. Can the authors expand on this?

* No statistical tests were described at any point. While they are not necessary for very striking functional data (such as 2c 3e,f), some assessment of confidence is required for interpreting the interaction energies from the simulation data.

Minor comments:

* Are there any differences in function between full-length TREK-2 and the TREK-2* construct?

* It is stated that lipid penetration through side fenestrations was observed but that this "did not obviously block ion conduction". As this has been suggested as an alternative mechanism for channel gating, it would be helpful to mark on Fig. 4c-d time points when lipid penetration occurs.

* Fig 2i does not appear to be mentioned in the text. On this topic: Can the authors comment on

why there appears to be voltage-dependent effects on the channel-membrane interaction energy in the Apo down-state in Fig 2i? Might this represent positive allosteric coupling between a flux-gated, conductive SF conformation and the "up-state"? Are differences of this magnitude meaningful?

* "In contrast, decyl-MTS modification caused a dramatic drop in NFX sensitivity (Fig. 3c)..." this appears to refer to Fig. 2c.

* Fig. 3j lower panels do not appear to be described in the text.

* The lack of 4 occupant K⁺ ions in the down state was originally described in Dong et al., 2015, which could be mentioned in discussion at, "we noticed that ion density at the S1 site was only present in the up-state structure..."

* Other very minor edits in the introduction should include: swapping "build" in place of "built up" before "a pseudo-tetrameric selectivity filter"; swapping "characterized by" in place of "characterize be."

Reviewer #1:

This is an interesting study of the molecular factors modulating the activity of the TREK-2 K²P potassium channels. Structural studies have revealed a up- and down-state conformation TREK-2. The up-state is believed to represent the highly active state induced by membrane stretching. But what is going on is unclear. In both states, the selectivity filter is essentially unchanged. The only structural changes in the selectivity filter have been only revealed at low potassium (K⁺) concentrations. The down-state is believed to be inactive (non-conductive). In some structures, a detergent or lipid molecules is observed blocking the pore, though it is unclear whether this is an artifact of purification (lie with the nAChRs channels). The stated goal of the study is to elucidate the role of the proximal C-terminus, its coupling to the selectivity filter, and the gating mechanism from the filter.

The results about the effect of phosphorylation promoting the up to down transition appear reasonable. In effect, the up and down states appear to be governed by changes in solvation free energy. Hydrophobic interactions promote the position in the membrane (up) and phosphorylation promote the position in the solution (down). The timescale of the simulations is probably too short to reveal the entire transition, but they move in the correct direction. This also point to a main weakness in the usage of MD here. It is all based on relatively mind-less brute-force simulations, with no attempt to characterize free energy differences using enhanced sampling methodologies (meta dynamics, umbrella sampling, etc). It is very nice to see that a piece of an helix is moving in the right direction, but it would be more convincing to show that the free energy landscape has been altered quantitatively. Furthermore, the analysis of the short-range Coulomb and Lennard-Jones energies between the pCt and the membrane with regards to phosphorylation is pretty meaningless. The hydrophobic effect is not a LJ interaction, and direct electrostatic interactions are all shielded by solvation (dielectric effect in the language of continuum electrostatics).

Answer: We sincerely appreciate the reviewer for their insightful suggestions. We fully concur with the reviewer's observation regarding the shortcomings of the unbiased MD simulations as presented in the previous version of the manuscript. Therefore, during the revision process, we performed additional sets of simulations including enhanced sampling simulations:

- (i) For the systems involving MTSET⁺ and decyl-MTS attachments, we employed adiabatic biased MD (ABMD) to facilitate the transition of the C-terminus from the down to up state. In this context, we characterized the forces required for this transition. Notably, our results showed that, as anticipated for the hydrophobic decyl-MTS system, less force was needed compared to the MTSET⁺ attached system (results shown in Fig. 2i, see below).
- (ii) We also extended our unbiased simulations of phosphorylated states from 1 μ s to 3 μ s, while simultaneously increasing the temperature from 300 K to 320 K. These simulations started when the C-terminus was in an up-state conformation and in-silico introduced the phosphorylation at the PKC site. In 3 out of 7 simulations, we observed a spontaneous full transition from the up-state to the down-state, accompanied by conformational changes in the SF during the simulations (Fig. 4g-j). Notably, in the non-conductive conformation of the down-state, the C-terminus could adopt a position even deeper than that observed in the X-ray down-state

conformation. This finding is in line with a prior computational study on the TREK-2 channel (PMID: 35022758). We believe these simulations, which demonstrate spontaneous full transitions in the C-terminus and simultaneous inactivation of the SF, provide strong evidence for the coupling between the C-terminus and the SF.

- (iii) We conducted well-tempered metadynamics simulations for both the up- and down-states, selecting the backbone dihedral angle changes of F284 and Y175 in the SF as two CVs. The free-energy surface (FES) profiles unveiled three energy minima for the up-state (Fig. 4n): (i) up-I corresponds to the conductive state, resembling the X-ray structure; (ii) up-II represents a single flipped conformer of F284; (iii) up-III, the lowest energy conformation, corresponds to a doubly flipped conformation of F284 and Y175. In contrast, for the down-state, we identified four energy minima. Down-I and down-II both correspond to single flipped conformations of Y175, while down-IV represents the single flipped conformation of F284. Down-III corresponds to a doubly flipped conformation of both F284 and Y1175. Notably, no energy minimum for the conductive state was observed in the down-state. This finding underscores a distinct energy difference between the conductive and non-conductive conformations in up- and down-state simulations. Additionally, we made extensive efforts to simulate the up-to-down transition in the C-terminus using well-tempered metadynamics, experimenting with different CVs and Gaussian heights. However, we encountered challenges in achieving satisfactory convergence, primarily due to the high flexibility of the C-terminus, a characteristic also highlighted in the introduction of manuscript. In response to these findings, we have incorporated a new secondary structure analysis in the revision, illustrating the flexibility of C-terminus throughout the simulations (Figure S20 and S21). Consequently, the C-terminus exhibits high degree of freedom, complicating the derivation of a reliable energy landscape with our current computational resources. We will defer this aspect to future studies using other enhanced sampling methods.

We agree with the reviewer that the analysis of the short-range Coulomb and Lennard-Jones energies between the pCt and the membrane with regards to phosphorylation is not very meaningful. Therefore, in the revised version of the manuscript, we have replaced these data with the results obtained from the ABMD (Figure 2i).

Fig. 2. Membrane tethering of the pCt with cysteine modifying probes activates TREK channels. (a) Example recordings of WT and F325C mutant TREK-1 channels in a symmetric K⁺ gradient at pH 7.4 at +40 mV showing no effect upon application of 100 μM decyl-MTS for WT and robust activation for F325C mutant channels. (b) Systematic screening of cysteine mutants in the pCt region of TREK-1 by recordings as in (a) reveal mutants that are activated (A) with decyl-MTS. Note, side chains of mutant channels that are strongly activated (green sticks) point to the inner leaflet of the lipid bilayer phase (gray sticks) as highlighted for TREK-2* in the inlay. Mutant TREK-1 channels with robust decyl-MTS activation show an inhibition (I) upon MTS-ET⁺ application. (c) Apparent affinities (IC₅₀ values) of NFx for WT, F325C mutant and decyl-MTS-activated F325C mutant TREK-1 channels. (d,e) Example recordings as in (a) of TREK-1 homologous mutant channels with the strongest decyl-MTS response in TREK-2* (d) and TRAAK (e). Note, TREK-2* for functional recordings corresponds to the construct used in MD. (f) Representative initial (left) and final (right) snapshots of decyl-MTS-modified TREK-2* from a 1 μs unbiased MD simulation. (g) A representative final snapshot of

MTS-ET⁺-modified TREK-2* from a 1 μ s unbiased simulation. **(h)** The distance (D_z component of the vector) between a chosen atom of the ligand (C12 for decyl-MTS, N2 for MTS-ET⁺) and the membrane plotted against time over 5 runs of unbiased 1 μ s simulations, conducted under +200 mV. The membrane was defined as a plane by selecting the center of geometry of phosphorus atoms located in the cytosolic part of the POPC lipid bilayer. **(i)** The average total force required for the down-to-up transition of M4/pCt for a decyl-MTS and MTSET⁺ attached TREK-2 channel, respectively, derived from 5 runs of 10 ns ABMD simulations. Green circles represent the total force for single runs.

Fig. 4. Phosphorylation-mediated state transition of TREK channels. (a) Cartoon depicting gating model and state dependency of NFx inhibition in TREK K_2P channels. (b) Example recording of WT TREK-1 channels in an asymmetric K^+ gradient at pH 7.4 in a

voltage range from -80 to +80 mV showing the dose-dependent inhibition with the indicated concentrations NFX (left) and the time course of block analyzed at +40 mV (right). (c) Same recording and analysis as in (b) for S315A TREK-1 channels. (d) Dose-response curves of NFX inhibition for WT and mutant TREK-1 channels as indicated analyzed from recordings as in (b). (e,f) NFX IC_{50} values from dose-response curves as in (d) for WT and mutant TREK-1 (e) and TREK-2 (f) K_{2P} channels. (g) Time course of the θ -angle for M4 in the apo and phosphorylated TREK-2* simulations starting from the up-state structure, shown for the chain A and B. Histograms of θ -angle distribution are included on the right site. θ -angle of the up-state and down-state X-ray structures was used as a reference (h) and shown as dashed line. Simulations with unstable SF are indicated with a red arrow. (i) Distribution of Psi angle (ψ) for F284 and Y175 from three replicates of 3 μ s phosphorylated simulations with unstable SF. (j) Traces of ions passing through the SF during a representative 3 μ s phosphorylated TREK-2* simulation. The temperature of the system is increased from 300 K to 320 K after 1000 ns. (k) The distance between S331 and F202 was illustrated for the unphosphorylated TREK-2* in the up-state and (l) for a phosphorylated TREK-2* after 1 μ s simulation. (m) Representation of selected collective variables (Ψ of F284 and Y175) for well-tempered metadynamics simulations. (n) Two-dimensional PMF derived from a 500 ns well-tempered metadynamics simulation for the up-state and down-state TREK-2*, respectively. The free energy surface contoured in steps of 4 kJ/mol up to 80 kJ/mol. Three energy minima were identified for the up-state, while four energy minima were identified for the down-state.

Fig. S20. Secondary structure analysis of up-state TREK-2*. Time-dependent evolution of secondary structure calculated using gromacs DSSP tool for 5 runs of 1 μ s up-state TREK-2* simulations.

Fig. S21. Secondary structure analysis of down-state TREK-2*. Time-dependent evolution of secondary structure calculated using gromacs DSSP tool for 5 runs of 1 μ s down-state TREK-2* simulations.

The least convincing aspect of the work concerns the ion conduction simulations. Simulations of the up-state at +200 mV displayed five K⁺ conduction events on average. But simulations at -200 mV did not conduct ions. Simulations of the down-state also did not conduct. Does one expect the IV curve to display such non-linear rectifying behavior for the active channel? Overall, these are very small and relatively inconsistent differences in behavior between these various systems. As a point of comparison, the dilated C-type inactivated Shaker channel from Swartz and co-workers (Tan et al, Science Advances 2022) conducts K⁺ in MD simulations unless the structure is restrained. The pathway along the dilated conformation is not occluded, so the free energy barrier must be just larger to stop conduction. One big concern is whether MD force fields are accurate enough to capture such a subtle inactivation mechanism. The main difference between the conductive and non-conductive selectivity filter is attributed to ion occupancy in the side S1 and the reorientation of the backbone carbonyls of Y175 and F284. But here again, the actual free energy difference dictating the conformational change is not characterized. Ultimately, one does not say why are these carbonyls affected by the rest of the structure. Could one predict one mutation that would prevent this change in the selectivity filter?

Answer: Experimentally K_{2P} channels exhibit voltage-dependent gating behavior, with our previous functional and mutational studies suggesting that the SF is responsible for this voltage-sensitivity (PMID: 26919430). However, MD simulations of other K⁺ channels using similar approaches, also show strong outward rectification (PMID: 25324389, 33602810), even though this is not the case experimentally. Therefore, we cannot definitely determine whether the lack of conductivity under negative voltage is directly linked to the voltage-sensitivity of the channel or if it results from force field imperfections or others. Consequently, we have decided not to directly associate this observation with voltage gating in the TREK channels. Further systematic studies on this behavior will be necessary in the future. We included now a

following discussion in the manuscript “*It is noteworthy that we observed very few ion conduction events under negative transmembrane potential for the up-state and no ion conduction for the down-state (Supplementary Fig. S6). Whether this result is related to voltage dependency of the TREK channel as previously revealed²⁷, or it is an artifact associated with force field imperfections as demonstrated in other K⁺ channels using similar methods^{42,43}, future systematic studies are warranted.*”. We agree with the reviewer that presenting the results of ion conduction simulations under negative voltage may potentially mislead readers. Therefore, in the revised version, we have moved this part of the results into the SI.

Nonetheless, we still believe the results from the ion permeation simulations for both the up- and down-states are of significant importance. These simulations have not only revealed differences in ion conductance but also unveiled robust conformational changes in the SF between the up- and down-states. In the up-state, the SF conformation maintains a conductive conformation in all five simulation runs, whereas in the down-state, conformational changes at the S1 site were consistently observed across all independent runs. It is also worth noting that the down-state of TREK-2 represents the most pronounced down-state structure among the various resolved TREK structures. A number of previous ion permeation simulations have primarily focused on the up-state of TREK-1/2 and TRAAK (PMID: 34323472, 30792303, 31772184), demonstrating that the channels remain conductive with a stable X-ray conformation throughout the simulations. In a simulation study of TREK-2, where membrane stretch was applied, the authors noted M4 helix (excluding the pCt) shifting towards the up-state, resulting in increased conductance (PMID: 29590591). We believe that our current study systematically addresses this issue. By incorporating the pCt into the channel model, we observed a significant difference in the interaction between the membrane and pCt, and how phosphorylation influences the transition from up to down.

In the revised manuscript, following the reviewer’s suggestion, we have included the results of the metadynamics simulations for both the up- and down-state, respectively, selecting the dihedral angles of Y175 and F284 as two CVs. Here, we observed a significant difference in the energy landscape between the simulations conducted for the up- and down-states, as discussed above.

Furthermore, from the extended simulations of the phosphorylated system, we observed complete transitions from up-to-down states, accompanied by conformational changes in the SF, especially involving the SF residues at the S1 binding site. Based on these simulations, we conducted a new interaction analysis using PSNtools, a graph-based computational method for analyzing protein interaction networks (PMID: 35140884). This approach allows us to understand allosteric communication between distal sites by computing the shortest communication pathways through the interaction network. As shown in Figure 5, communications between the C-terminus in the up-state and SF in the conductive conformation primarily occurs through the M4-P2-SF pathway, whereas in the down-state when the SF adopts a non-conductive conformation, this network shifts to the M4-M4 loop-SF pathway. In the revised manuscript, we attempted to make this point to be more clear.

We have indeed identified a number of key interactions that mediate the communication between the pCt/M4 and the SF. Nonetheless, this process is complex, characterized by the competition between two pathways: (i) M4 → P2 → SF and (ii) M4 → M4 loop/EC loop → SF.

Furthermore, systematically evaluating the effects of individual mutations requires substantial computational resources and time, exceeding what is currently available to us. Therefore, we have decided to leave this aspect open for future study.

Fig. 5. Coupling mechanism between pCt/M4 and SF gate. (a) Analysis of the key interaction differences between the up and down states revealed that the dynamics of pCt/M4 are coupled with the inter- and intra-chain interactions between M2 and M4. These, in turn, are linked to the SF gate via two distinct pathways: (i) M4 → P2 → SF; (ii) M4 → M4 loop/EC loop → SF. Favorable interactions in the up-state are depicted with red dashed lines, while blue dashed lines indicate favorable interactions in the down-state. **(b)** Network analysis using PSNtool (ref) of the phosphorylated simulations, which showed a complete transition from up to down in the C-terminus, confirmed that the shortest communication path from the pCt/M4 to the SF was mediated by the P2 helix (left/middle). In the down-state, the network shifted to follow the M4 → M4 loop/EC loop → SF pathway (right).

In final analysis, while the combination of experiments and computations is highly commendable, I find that the main results are suggestive, but not entirely compelling. Some trends can be explained qualitatively (effect of phosphorylation for example) but would benefit

from more systematic enhanced sampling methodologies. Other results such as the conduction or lack of conduction of the channel in the up or down states display small differences of uncertain statistical significance and proposed molecular explanations have to rely on subtle effects that may be highly dependent on the force fields and sampling.

Answer: We agree with the reviewer's assessment that the previous version of the manuscript, which solely included results from unbiased MD simulations, was not conclusive enough. In response to the reviewer's suggestions, we conducted two new series of enhanced sampling simulations and extended the simulations of the phosphorylated system. These simulations have now allowed us to observe the complete transition from the down to up state, as well as to compare the free energy landscape of the SF inactivation between up- and down-state. We are aware that the simulations performed with an additive force field may exhibit inaccuracies in certain cases. However, our results are mainly based on the comparison of the simulations performed for up- and down-state as well as phosphorylated system, where the same method and simulation protocol was employed. Here, we observed a large difference in the conformational behavior in the SF. Furthermore, as noted by the reviewer, these simulations are strongly supported by a high number of functional data (activity difference in electrophysiology and previous gating charge analysis) and structural (X-ray structures with low K⁺ condition showing similar ion depletion at S1 and S2 binding sites). Therefore, we believe that our manuscript is now in a significantly improved position to elucidate our originally proposed mechanism. Consequently, we decided to incorporate a new paragraph in the discussion section to discuss the issue of force field inaccuracy as follows: "*Nevertheless, we propose that the configuration without ions at the S1 site, coupled with the carbonyl reorientation of Y175 and F284, does not represent the true SF conformation of inactivated TREK-2^{*} channels. Previously, from the gating charge analysis, we proposed the inactive filter to be in an 'ion-deplete' state²⁷. The full inactivation of the SF was not observed in the current study most probably due to the limited time scale of the simulations, as well as the choice of using additive force field — without considering the polarization effect — may contribute to additional inaccuracy. Nonetheless, based on the strong functional and simulation evidences discussed above, we are confident to provide a strong link between the dynamics of pCt/M4 and C-type gating in the SF.*"

Minor points:

A criticism concerns the style. Many sentences use "functional electrophysiology" to mean "experimental" measurements, and "computational electrophysiology" to mean MD simulations. I had to re-read carefully several statements to make sure if it was about an experimental fact or a MD result. Personally I dislike the expression computational electrophysiology, which I find needlessly confusing. It is nice to coin cute names, but this does not help the clarity of the text. In addition, there are unjustified emphatic affirmations throughout the text. For example page 12: "Our findings so far have established that the upward movement of the pCt induces a highly active channel state that likely resembles the crystallographic up-state". Highly active? How? Or page 17: "Microsecond MD simulations of apo up- and down-state TREK-2 with extended pCt revealed a striking difference in the conductance of two distinct conformational states" So now the difference of 5 ions conducted has become "striking"?

Answer: We thank the reviewer for these useful suggestions. We replaced now “computational electrophysiology” by “MD simulations”. We rephrased the sentences mentioned by the reviewers as follows. On page 12: “Our findings so far have established that the upward movement of the pCt induces an active channel state that likely resembles the crystallographic up-state.” On page 17: “Microsecond MD simulations of apo up- and down-state TREK-2 with extended pCt revealed a notable difference in the conductance of two distinct conformational states.”

The transmembrane potential was implemented via the charge imbalance across a double bilayer system. The simpler and more direct alternative would have been to implement the transmembrane potential via an external linear electric field. With the charge imbalance methodology, the transmembrane potential has to be tightly monitored to show that it is indeed the desired target value. The issue is even more critical in the case of non-equilibrium ion conduction. It is stated "An ion difference of 2 was introduced between the two compartments separated by lipid bilayers to introduce a transmembrane potential of about ± 200 mV. Additional particle interchange algorithm (deterministic protocol) were employed during MD simulations to keep the number of ions in every compartment constant alchemically." This needs to be better documented.

Answer: We think that both the double bilayer setup method for maintaining the transmembrane potential and the use of an external linear electric field have their respective advantages and disadvantages. The external linear electric field can be easily adjusted to the desired target value, whereas the double bilayer setup more closely resembles biological cells, where the ionic gradient across the cell membrane is the driving force for ion conduction. Furthermore, we think the bilayer setup is less affected by artifacts introduced by periodic boundary conditions compared to the linear electric field method. Nonetheless, a number of MD studies on other potassium channels, such as KcsA and MthK, conducted in Bert de Groot’s lab using both methods, showed no noticeable differences in the ion conduction mechanism.

Heeding the reviewer's suggestion, we calculated the electrostatic potential profiles for the up- and down-state TREK-2 simulations across the simulation box and derived the standard deviation from five 1 μ s simulation runs. This analysis enables a more accurate assessment of potential fluctuations for each simulation setup. The figure is now included as Figure S18 in SI.

Fig. S18. Transmembrane potentials during the MD simulations. Electrostatic potential along the z-axis for up- and down-state TREK-2^{*} simulations. Electrostatic potentials were derived from 5 runs of 1 μ s simulations with an ion imbalance of 2 between two compartments in the CompEL MD setup.

In some of the simulation runs a lipid occupied a site between M2 and M4 (Fig. 3L). Are there two lipids associated with the channel (in a symmetric manner) or only one lipid is observed?

During the simulations, we observed multiple lipids entering the pore region. The exact number of lipids fluctuates with the time. We included now the data in the Fig. S7.

Fig. S7. Number of lipids occupying the pore during up- and down-state simulations. Number of POPC lipids occupying the pore (defined in the analysis as a cylinder) for (a) up- and (b) down-state TREK-2 simulations over the course of the simulation time. In the rep2 of up-state simulations, no lipid entered the pore. The center of the cylinder is positioned 10 Å below the center of mass of the selectivity filter residues T171 and T280, located near the S4 ion-binding site. The height of the cylinder is defined 40 Å and radius is defined 8 Å.

Reviewer #2:

The gating mechanism of two-pore domain potassium channels is still far from being completely understood. This study reports novel information about how the C-terminal of helix M1 regulates the transition between the down- and up-states of the channel and how the movements of this C-terminal modify the properties of the selectivity filter. These results are interesting contributions to the current understanding of how phosphorylation regulates TREK channels. The methods are thoughtfully planned, with experiments and molecular dynamics simulations that coherently support the conclusions of the manuscript. In summary, I believe that this is a well-designed and clearly written work that significantly contributes to the field. Below, I have included some minor comments.

- The Methods section does not report the initial configuration of ions and water molecules inside the selectivity filter and cavity of the channel. Given the low number of conduction events (null in some atomic models), it is not possible to exclude that some results might depend on the initial ion configuration inside the channel. Therefore, I suggest the authors to explicitly discuss this issue in the manuscript, clearly define how and why the atomic models were initialized with respect to the ions inside the channel, and confirm if this initialization was consistent across all replica.

Answer: We thank the reviewer for this helpful suggestion. We have now included the following information in the Methods section: The simulations were initiated with a four-ion configuration (S1-S4) in the SF for all setups, consistent with the ion configuration in the X-ray structure of the TREK channel in the up-state (PDB ID: 4BW5). We used the same ion configuration in the down-state simulations to avoid any influence on ion conductance caused by different initial ion configurations.

In agreement with the discussion in the final part of page 18, the depletion of ions from S1 was recently reported in simulations of the hERG channel by my group in *J. Chem. Inf. Model.* 2023, 63, 251–258, where we formulated a similar hypothesis that this event might represent an initial step of inactivation.

Answer: We greatly appreciate the referee highlighting this important reference on hERG channel. Notably, the hERG channel, similar to the TREK channels, also features a Phe instead of a Tyr at the S1 binding site. These findings suggest that hERG and TREK channels may share a very similar C-type inactivation mechanism. We have cited this work in the discussion section on page 18 as follows: Interestingly, a recent MD work by Pettini et al. also revealed a similar depletion of ion binding at S1, which is proposed to present an initial step of inactivation...The hERG (human Ether-a-go-go Related Gene) channels, which also possess a Phe at the homologous position in the filter, exhibit side-chain shifts in the cryo-EM structures and ion depletion at S1 in the MD simulations, which were suggested to be associated with fast inactivation.

- These are a few typos or unclear sentences that I noted: page 3 “membrane leaflet with opening”, “recently cryo-electron”; page 4 “activation AA and anesthetics”; page 9 “appeared to dominated”; page 17 “interaction differences in interactions”.

Answer: These mistakes have been corrected.

Reviewer #3:

This study by Türkaydin and colleagues combines functional studies of TREK channels with MD simulations and proposes an explanation for the molecular mechanism of phosphorylation-mediated regulation of channel activity, and a general framework for how down-to-up transitions of TM4 is coupled to activation of the SF gate. Using elegant MTS-modification, phospho-mimetic mutagenesis, and norfluoxetine sensitivity experiment - and comparison with MD simulations - the authors convincingly show that membrane anchoring of TM4 increases channel opening which is a plausible explanation for the inverse inhibitory effect of phosphorylation of the extended M4 helix.

The molecular dynamics approaches are mostly outside of my expertise and so I cannot comment on the technical elements. However, the interaction network analysis supports previous models that upwards movement of TM4 is associated with opening of the SF gate and highlights previously identified and potential novel residues responsible.

The paper is very well written, experiments are beautifully performed and displayed, and results provide interesting advances which improve our understanding of TREK channel gating.

We thank the reviewer for the positive evaluation of our work and the many helpful comments to improve the manuscript. The reviewer specific comments are addressed below:

However, the model could be tested, validated, and strengthened significantly by the following - and this is an excellent opportunity to clarify some important outstanding questions in the field:

Data suggests phosphorylation of TREK channels drives conversion to low activity “down”-state. This is an interesting and important finding, which would help to explain TREK channel regulation in vivo. The (de)phospho-mimetic experiments are illuminating, but a telling killer experiment showing direct phosphorylation effects is lacking. For example, can the authors excise patches from oocytes in which the PKC pathway has been inhibited, and then demonstrate that NFX sensitivity is markedly reduced, as would be expected for their model? This would provide strong support for a phosphorylation-induced shift in up vs down conformations.

Answer: We can only concur with the reviewer here that a direct intervention in the phosphorylation pathway via inhibition of PKC and subsequently testing with norfluoxetine would be at least as informative, if not even superior, to the (de)phosphorylation mimics. We have conducted the suggested experiment with oocytes injected with WT hTREK-2 cRNA and incubated them for 48 h at 16.9 °C. Following this, we directly incubated the oocytes for additional 4 h in standard OR2 solution containing 50 μM of the phosphokinase C (PKC) inhibitor Bisindolylmaleimide I (BIM I) and measured NFX sensitivity of hTREK-2 in excised patches of these oocytes. The affinity of WT TREK-2 for NFX after incubation was not altered ($IC_{50} = 2.1 \pm 0.4$, $n = 6$) compared to TREK-2 of untreated oocytes ($IC_{50} = 2.7 \pm 0.4$, $n = 31$). However, we also did not observe a current increase upon direct application of BIM I.

Further, we also tested the direct application of 1.0 μM of the PKC inhibitor BIM I to HEK293 cells (in whole-cell configuration) expressing hTREK-1. Similarly, under these conditions, TREK-1 channel currents were not altered within 30 min. ($n = 6$). Thus, it appears that under these experimental condition PKC inhibition does not result in a detectable dephosphorylation of the PKC site. Unfortunately, it is not known which phosphatase mediates the dephosphorylation of the PKC site in TREK channels nor are pharmacological phosphatase activators known that could be used promote the dephosphorylation. Thus, we did not pursue this any further.

The interaction network analysis provides possible explanations for how TM4 movement is transduced to SF gate opening. As mentioned, this analysis highlights a number of residues which have previously been shown to significantly affect channel activity when mutated (e.g. W326, R237 in Dong et al., 2015). The authors are well placed to determine whether mutation of these residues, *in silico*, destabilizes the down-state and to show how this might promote stabilization of the conductive SF.

Answer: We agree with the reviewer's suggestion that simulating various mutants, identified through interaction network analysis and previous mutation studies, can aid in studying the conformational change from down- to up-state in the channel and how this is related to the conformational change in the SF. However, the timescale for the conformational movement in the C-terminus is relatively long, as demonstrated in our simulations of phosphorylated TREK-2 channels. In only 3 out of 7 simulations, each lasting 3 μs , did we observe a complete transition from up to down. Hence, unbiased molecular dynamics (MD) simulations are not well-suited for assessing the impact of mutations. During the revision, we also explored whether we could characterize the transition from up to down, or vice versa, using enhanced sampling methods, as suggested by the first reviewer. Nonetheless, our metadynamics simulations failed to achieve satisfactory convergence due to the high flexibility of the C-terminus, which results in a high degree of freedom. We, therefore, have to leave this aspect for future investigation.

The interaction analysis proposes novel functionally important residues. Patch clamp analysis of key mutants would significantly strengthen the proposed model. For example, does mutation of C189, the G208/L211/G212 cluster, T278, or T280 affect channel opening, up-state stability, phosphorylation-regulation or NF κ B sensitivity?

Answer: We agree with the reviewer that functional investigation of key residues highlighted by our interaction network analysis would significantly strengthen the proposed gating model. Indeed, we performed a systematic unbiased mutagenic perturbation analysis of TREK-1 and TREK-2 K_{2P} channels to identify residues that are critical for the stability of the down and up state, respectively. This analysis included more than 80 positions in TM2, TM3 and TM4 where we determined the effect of mutagenic exchange on the relative P_o as well as on the NF κ B sensitivity and is complemented with extensive equilibrium and non-equilibrium MD simulations of all critical mutations. In the context of this study we also investigated the residues the reviewer suggested such as C189, G208, L211, G212, T280, W326 and R237 (see Figure). Indeed, consistent with our interaction network analysis mutation at these positions strongly increased the channel P_o and also markedly reduced the NF κ B sensitivity indicating their involvement in our proposed gating pathway. However, in addition to the down-to-up-state gating pathway, we also identified several mutations that resulted in a high P_o

without changing the NFx sensitivity suggesting an additional down-state activation pathway that is unrelated to the phosphorylation induced gating mechanism. Because of the complexity of the mutagenic perturbation analysis, we decided to publish these results in a separate manuscript, however, the findings of this manuscript are in full agreement with the gating mechanism that we propose to underly the regulation of TREK activity by phosphorylation. Unfortunately, the second manuscript is not citeable as a preprint yet but we would like to show the reviewer one of the preliminary figures of the result section showing P_O and NFx sensitivity of key residues that we also identified in the contact analysis of this manuscript.

Strikingly, mutations perturbing the down-state network (e.g., residues of the so called `triade` interaction site including E223 (TM2), R237 (TM3) and W326 (TM4) or residues of the intra-chain `fenestration` cluster including A318 (TM4), G208 (TM2), L211 (TM2), G212 (TM2)) exhibit an increased relative P_O at rest and a distinct reduced NFx sensitivity indicating that channels adopted a more up-state-like conformation.

Major comments:

Despite the interaction network analysis, it was not clear to me why the conductive SF conformation is disfavored in the down conformation. Is there a temporal correlation with CO-flipping and key interactions elsewhere in the channel. Can the authors expand on this?

Answer: We acknowledge the reviewer's observation that the earlier version of our manuscript did not adequately address this point. In this revised version, we included the metadynamics simulations to characterize free energy differences of the conformational changes in the SF between up- and down-states. In the well-tempered metadynamics simulations for both the up- and down-states, selecting the backbone dihedral angle changes of F284 and Y175 in the SF as two collective variables (CVs). The free-energy surface (FES) profiles unveiled three energy minima for the up-state (Fig. 4n): (i) up-I corresponds to the conductive state, resembling the X-ray structure; (ii) up-II represents a single flipped conformer of F284; (iii) up-III, the lowest energy conformation, corresponds to a doubly flipped conformation of F284 and

Y175. In contrast, for the down-state, we identified four energy minima. Down-I and down-II both correspond to single flipped conformations of Y175, while down-IV represents the single flipped conformation of F284. Down-III corresponds to a doubly flipped conformation of both F284 and Y1175. Notably, no energy minimum for the conductive state was observed in the down-state. This finding underscores a distinct energy difference between the conductive and non-conductive conformations in up- and down-state simulations.

Additionally, during the revision we extended our unbiased MD simulations to 3 μ s and increased the temperature from 300 K to 320 K, an approach that is often used to accelerate dynamics in the MD simulations. In 3 out of 7 simulations, we observed a complete up-to-down transition and a corresponding conformational change at the S1 binding site in the SF (Fig. 4g-j). Here, we noted the C-terminus could adopt an even deeper down-state compared to the original X-ray structure when the SF inactivates. Based on these observations, we conducted a new interaction analysis using PSNtools, a graph-based computational method for analyzing protein interaction network (PSNtools, PMID: PMID: 35140884). This approach allows us to understand allosteric communication between distal sites by computing the shortest communication pathways through the interaction network. As shown in Figure 5, communications between the C-terminus and the SF in the up-state are mainly mediated through the M4-P2-SF pathway, whereas in the down-state when the SF inactivates, this network shifts to the M4-M4 loop-SF pathway. This network analysis again shows that the conductive conformation of the SF and the up-state of the M4/pCt is mainly stabilized by a number of interchain interactions between M2 and M4, M2 and P2 helix and P2 helix and the SF, while the carbonyl flipped conformation of the SF and the down-state of the M4/pCt is stabilized by a series of intra-chain interactions between M2, M3 and M4, as well as M4 loop and the SF.

Fig. 4. Phosphorylation-mediated state transition of TREK channels. (a) Cartoon depicting gating model and state dependency of NFx inhibition in TREK K_2P channels. **(b)** Example recording of WT TREK-1 channels in an asymmetric K^+ gradient at pH 7.4 in a

voltage range from -80 to +80 mV showing the dose-dependent inhibition with the indicated concentrations NFX (left) and the time course of block analyzed at +40 mV (right). **(c)** Same recording and analysis as in (b) for S315A TREK-1 channels. **(d)** Dose-response curves of NFX inhibition for WT and mutant TREK-1 channels as indicated analyzed from recordings as in (b). **(e,f)** NFX IC_{50} values from dose-response curves as in (d) for WT and mutant TREK-1 (e) and TREK-2 (f) K_{2P} channels. **(g)** Time course of the θ -angle for M4 in the apo and phosphorylated TREK-2* simulations starting from the up-state structure, shown for the chain A and B. Histograms of θ -angle distribution are included on the right site. θ -angle of the up-state and down-state X-ray structures was used as a reference **(h)** and shown as dashed line. Simulations with unstable SF are indicated with a red arrow. **(i)** Distribution of Psi angle (ψ) for F284 and Y175 from three replicates of 3 μ s phosphorylated simulations with unstable SF. **(j)** Traces of ions passing through the SF during a representative 3 μ s phosphorylated TREK-2* simulation. The temperature of the system is increased from 300 K to 320 K after 1000 ns. **(k)** The distance between S331 and F202 was illustrated for the unphosphorylated TREK-2* in the up-state and **(l)** for a phosphorylated TREK-2* after 1 μ s simulation. **(m)** Representation of selected collective variables (Ψ of F284 and Y175) for well-tempered metadynamics simulations. **(n)** Two-dimensional PMF derived from a 500 ns well-tempered metadynamics simulation for the up-state and down-state TREK-2*, respectively. The free energy surface contoured in steps of 4 kJ/mol up to 80 kJ/mol. Three energy minima were identified for the up-state, while four energy minima were identified for the down-state.

Fig. 5. Coupling mechanism between pCt/M4 and SF gate. (a) pCt/M4 dynamics trigger different inter- and intra-chain M2 - M4 interactions that are further coupled to the SF gate via two different pathways: (i) M4 → P2 → SF; (ii) M4 → M4 loop/EC loop → SF. Favorable interactions in the up-state are depicted with red dashed lines, while blue dashed lines represent favorable interactions in the down-state. **(b)** Protein structure network analysis using PSNTool on a 3 μ s phosphorylated TREK-2 simulation confirmed that the up-state (the initial 1 μ s) is mainly stabilized by the M4 → P2 → SF pathway, whereas in the second 1 μ s during the transition between up- and down-state and SF inactivation, the interaction pathway shifted to M4 → M4 loop → SF.

No statistical tests were described at any point. While they are not necessary for very striking functional data (such as 2c 3e,f), some assessment of confidence is required for interpreting the interaction energies from the simulation data.

Answer: We agree with the reviewer that applying statistical tests to our simulation data would enhance the confidence in interpreting some of the results. However, in this study, due to the relatively large size of the systems that require extensive simulation time, we were limited to conducting only five to seven independent simulation runs for each setup. According to the checklist and recommendations provided by Nature Journal (PMID: 36918708), a minimum of three simulations per condition is advisable. Given that we only have data from these limited simulation runs, conducting a p-test is challenging. Nonetheless, as we noted in the main text, in all 5 simulations of the down-state, we observed a robust conformational change at the S1 binding site. Conversely, in all five up-state simulations, the SF remained a conductive conformation. Additionally, following the suggestion by reviewer 1, we have omitted the interaction energy data from our interpretation.

Minor comments:

Are there any differences in function between full-length TREK-2 and the TREK-2* construct?

Answer: This is indeed a valid question; to the best of our knowledge, TREK-2 and TREK-2* do not exhibit differential behaviour. The TREK-2* construct shows slightly reduced channel activity at resting state (pH 7.4) that is in line with previous findings of other N- and/ or C-terminus truncated TREK constructs (Dong *et al.* 2015, Lolicato *et al.* 2017). TREK-2* can be activated by low intracellular pH (pH_i 5.0), mechanical stretch and arachidonic acid comparable to WT. Further, we have investigated the NFx sensitivity in detail (TREK-2 WT IC₅₀ = 2.4 ± 0.7 μM, TREK-2* IC₅₀ = 3.2 ± 1.0 μM), in which both constructs do not show any remarkable differences.

It is stated that lipid penetration through side fenestrations was observed but that this “did not obviously block ion conduction”. As this has been suggested as an alternative mechanism for channel gating, it would be helpful to mark on Fig. 4c-d time points when lipid penetration occurs.

Answer: We agree with the reviewer and have indicated the time points when lipid penetration occurs on Fig. 4c,d. We also included a new analysis showing the number of lipids occupying the pore over the course of the simulation time (Fig. S7).

Fig. 3. Conductance, SF ion occupancy and conformational changes determined from TREK-2* simulations. (a) Ion conductance derived from apo TREK-2* simulations in its up- (PDB ID: 4BW5)¹⁴ and down-state (PDB ID: 4XDJ)¹⁴. Filled and empty circles represent mean and individual ion conductance, respectively. (b) One- and two-dimensional ion occupancy profiles within the SF for combined up-state and non-conductive down-state TREK-2* simulations. The radial area of the pore was defined and ions passing along the pore axis (D_z) were calculated from the simulations. The occupancy of ions was normalized per 0.001 \AA^3 per 1 \mu s based on the volume change along the radius. The center of mass of the SF backbone atoms was located on the 7 \AA point along the pore axis. Ion binding sites along the pore axis were indicated with dashed lines. (c) Traces of ions passing through the SF during a representative 1 \mu s up-state and (d) down-state TREK-2* simulation. Black arrows indicate the respective ion permeation events, while red arrows indicate lipid penetration from the lateral side fenestration. (e) Distribution of Psi angle (ψ) for F284 and Y175 from the combined up-state (left) and down-state (right) simulations. (f) Psi angles (ψ) variations of F284 in chain A and Y175 in chain B over time in the up-state (top) and down-state (bottom) of TREK-2*. Individual simulation runs were carried out with AMBER99sb⁴⁴ for 1 \mu s under a transmembrane voltage around $+200 \text{ mV}$.

Fig. S7. Number of lipids occupying the pore during up- and down-state simulations. Number of POPC lipids occupying the pore (defined in the analysis as a cylinder) for (a) up- and (b) down-state TREK-2 simulations over the course of the simulation time. In the rep2 of up-state simulations, no lipid entered the pore. The center of the cylinder is positioned 10 Å below the center of mass of the selectivity filter residues T171 and T280, located near the S4 ion-binding site. The height of the cylinder is defined 40 Å and radius is defined 8 Å.

Fig 2i does not appear to be mentioned in the text. On this topic: Can the authors comment on why there appears to be voltage-dependent effects on the channel-membrane interaction energy in the Apo down-state in Fig 2i? Might this represent positive allosteric coupling between a flux-gated, conductive SF conformation and the “up-state”? Are differences of this magnitude meaningful?

Answer: Reviewer 1 has pointed out a weakness in calculating the interaction energy between the C-terminus and the membrane. We concur with Reviewer 1 and have thus removed the

corresponding section. In its place, we have incorporated calculations of the forces required to transition the C-terminus from the down to up state in both decyl-MTS- and MTSET-attached systems, using adiabatic biased MD (ABMD) simulations. While voltage dependency is a very interesting aspect of K_{2P} channels, we are cautious about drawing any conclusions related to voltage gating in the current study. Some computational studies on other ion channels, which used a similar methodology, also displayed strong outward rectification in simulations (PMID: 25324389, PMID: 33602810), even though the voltage-dependency observed in experiments was substantially different. Therefore, we cannot definitely determine whether the lack of conductivity under negative voltage is directly linked to the voltage-sensitivity of the channel or if it is the result of other artifacts. Consequently, we have decided not directly associate this observation with voltage gating in the TREK channels, and further systematic studies on this behavior will be necessary in the future. We think that presenting the simulations under negative voltages may potentially mislead readers. Therefore, in the revised version, we have moved this part of the results into the SI and included a discussion in the main text as “*It is noteworthy that we observed very few ion conduction events under negative transmembrane potential for the up-state and no ion conduction for the down-state (Supplementary Fig. S6). Whether this result is related to voltage-dependency of the TREK channel as previously revealed²⁷, or it is an artifact associated with force field imperfections as demonstrated in other K^+ channels using similar methods^{42,43}, future systematic studies are warranted.*”

“In contrast, decyl-MTS modification caused a dramatic drop in NFX sensitivity (Fig. 3c)...” this appears to refer to Fig. 2c.

Answer: This mistake has been corrected.

Fig. 3j lower panels do not appear to be described in the text.

Answer: Thanks for noticing this mistake. The citation of the Figure 3j has been added now.

The lack of 4 occupant K^+ ions in the down state was originally described in Dong et al., 2015, which could be mentioned in discussion at, “we noticed that ion density at the S1 site was only present in the up-state structure...”

Answer: As suggested, we rephrased the text as “*Although no conformational difference was observed in the SF of the original X-ray structures between the up- and down-state TREK-2, we noticed that continuous ion density at the S1-S4 sites was present only in the up-state structure. In contrast, the down-state X-ray structure lacks ion density at site S1.*”

Other very minor edits in the introduction should include: swapping “build” in place of “built up” before “a pseudo-tetrameric selectivity filter”; swapping “characterized by” in place of “characterize be.”

Answer: The suggestions have been implemented.

Reviewers' Comments:

Reviewer #1:

Remarks to the Author:

The revised manuscript is certainly improved. My overall impression has not changed significantly. I feel that the computational results are suggestive, but continue to be not very compelling. In part, this is not entirely the fault of the authors because the channel is a large macromolecule that undergoes complex conformational changes affecting its function over timescales that challenge the accessible simulation time. But in part, some issues arise because the authors are using computational strategies and methodologies that are not the best. The adiabatic biased MD dates from 1999 [Marchi and Ballone, J. Chem. Phys. 110, 3697–3702 (1999)], and despite the striking ABMD acronym, it is essentially just an integration of the force along a steered MD trajectory. This is not a modern state-of-the-art enhanced sampling method by any sense. Increasing the temperature from 300 K to 320 K is throwing the kitchen sink at the sampling problem. The same can be said about the double bilayer charge imbalance to simulate ion conduction under a membrane potential. The issue is not about the average potential but about its time dependent: the passage of one ion causes an immediate jump in the membrane potential a single charge across the membrane changes the potential significantly (this is hidden is one only considers time averages). Thus, the double bilayer with charge imbalance is by no means "more closely resembles biological cells, where the ionic gradient across the cell membrane is the driving force for ion conduction" as the potential used to get IC curves is not generated from an ionic gradient the electrophysiological experiments. Efforts to simulate the up-to-down transition in the C-terminus using metadynamics failed, unsurprisingly. The confusing results about ion conduction are cleaned up by removing the annoying result of no conduction in the conductive state of the filter at negative potential. The authors worked hard with their simulations, but the methodologies are over the place and I am skeptical of many of the findings. I am all in favor of a synergy between experiments and computations, but the bar is very high to be able to claim that "new knowledge" about a channel in the real world has been produced by simulations. Are there suggestive observations in here, sure! But it is all about the general tone and how the results are communicated to the community. The danger is that real damage is done when presenting results as if they more solid than they are, misleading the scientific community.

Reviewer #2:

Remarks to the Author:

I've no further comment

Reviewer #3:

Remarks to the Author:

I thank the authors for a comprehensive and informative response. With the caveat that I do not have much experience with MD methods and would defer to the other reviewers here, I think the revision includes key additional details. I also appreciate the new patch clamp data and PKC inhibition attempts and have no further concerns (minus one or two typos which can be corrected in proofs).

I congratulate the authors on an elegant and insightful study, which is a significant advance for the field.

REVIEWER COMMENTS

Reviewer #1 (Remarks to the Author):

The revised manuscript is certainly improved. My overall impression has not changed significantly. I feel that the computational results are suggestive, but continue to be not very compelling. In part, this is not entirely the fault of the authors because the channel is a large macromolecule that undergoes complex conformational changes affecting its function over timescales that challenge the accessible simulation time. But in part, some issues arise because the authors are using computational strategies and methodologies that are not the best. The adiabatic biased MD dates from 1999 [Marchi and Ballone, J. Chem. Phys. 110, 3697–3702 (1999)], and despite the striking ABMD acronym, it is essentially just an integration of the force along a steered MD trajectory. This is not a modern state-of-the-art enhanced sampling method by any sense. Increasing the temperature from 300 K to 320 K is throwing the kitchen sink at the sampling problem. The same can be said about the double bilayer charge imbalance to simulate ion conduction under a membrane potential. The issue is not about the average potential but about its time dependent: the passage of one ion causes an immediate jump in the membrane potential a single charge across the membrane changes the potential significantly (this is hidden is one only considers time averages). Thus, the double bilayer with charge imbalance is by no means "more closely resembles biological cells, where the ionic gradient across the cell membrane is the driving force for ion conduction" as the potential used to get IC curves is not generated from an ionic gradient the electrophysiological experiments. Efforts to simulate the up-to-down transition in the C-terminus using metadynamics failed, unsurprisingly. The confusing results about ion conduction are cleaned up by removing the annoying result of no conduction in the conductive state of the filter at negative potential. The authors worked hard with their simulations, but the methodologies are over the place and I am skeptical of many of the findings. I am all in favor of a synergy between experiments and computations, but the bar is very high to be able to claim that "new knowledge" about a channel in the real world has been produced by simulations. Are there suggestive observations in here, sure! But it is all about the general tone and how the results are communicated to the community. The danger is that real damage is done when presenting results as if they more solid than they are, misleading the scientific community.

We are very thankful to Reviewer for pointing out that the suggestive nature of the findings from the MD simulations was not adequately reflected in the previous version. We agree that more advanced enhanced sampling methodologies should be explored in the near future to address the up-to-down transition and its coupling to the SF in a quantitative manner. In this revised manuscript, we have carefully reviewed our manuscript and revised the wording to be more cautious regarding the conclusions drawn from the MD results. All changes are highlighted in the manuscripts. Moreover, we have included a new paragraph in the conclusion section, specifically addressing the limitations and suggestive nature of our MD findings, as follows:

In conclusion, using an integrated approach, combining electrophysiology including systematic cysteine scanning mutagenesis, lipid tethering and blocker experiments, molecular dynamics simulations and interaction network analysis, we have highlighted the essential role of the pCt in TREK channel gating. We also revealed how the interaction between the pCt

domain and the membrane influence the conformational equilibrium between the up- and down-state of TREK channels. Furthermore, MD simulations proposed an allosteric mechanism at the atomistic scale, delineating two pathways of the energetic coupling between the cytosolic sensing domain and the SF. These pathways are in a good agreement with previous functional and mutational studies. However, it is important to note that while MD simulations at higher temperatures suggested the possibility for a complete conformational transition from the up- to down-state upon phosphorylation at the PKC site, our enhanced sampling simulations did not achieve sufficient convergence to quantitatively characterize free energy differences between these two states. This issue is most probably due to the high flexibility of the C-terminus (Supplementary Fig. S20, S21). Therefore, future mutational work is necessary to further validate the coupling pathways proposed in the current study. Additionally, more advanced enhanced sampling methodologies should be explored in the near future to better assess the conformational transition between the up- and down-states in a more quantitative manner.

Reviewer #2 (Remarks to the Author):

I've no further comment

Reviewer #3 (Remarks to the Author):

I thank the authors for a comprehensive and informative response. With the caveat that I do not have much experience with MD methods and would defer to the other reviewers here, I think the revision includes key additional details. I also appreciate the new patch clamp data and PKC inhibition attempts and have no further concerns (minus one or two typos which can be corrected in proofs).

I congratulate the authors on an elegant and insightful study, which is a significant advance for the field.